# COMBATING EXACERBATED HETEROGENEITY FOR ROBUST MODELS IN FEDERATED LEARNING

**Jianing Zhu**[1]   **Jiangchao Yao**[2,3†]   **Tongliang Liu**[4]   **Quanming Yao**[5]
**Jianliang Xu**[1]   **Bo Han**[1†]

[1]Hong Kong Baptist University [2]Shanghai Jiao Tong University [3]Shanghai AI Laboratory
[4]Sydney AI Centre, The University of Sydney [5]Tsinghua University

{csjnzhu, xujl, bhanml}@comp.hkbu.edu.hk Sunarker@sjtu.edu.cn
tongliang.liu@sydney.edu.au qyaoaa@tsinghua.edu.cn

## ABSTRACT

Privacy and security concerns in real-world applications have led to the development of adversarially robust federated models. However, the straightforward combination between adversarial training and federated learning in one framework can lead to the undesired robustness deterioration. We discover that the attribution behind this phenomenon is that the generated adversarial data could exacerbate the data heterogeneity among local clients, making the wrapped federated learning perform poorly. To deal with this problem, we propose a novel framework called *Slack Federated Adversarial Training* (SFAT), assigning the client-wise slack during aggregation to combat the intensified heterogeneity. Theoretically, we analyze the convergence of the proposed method to properly relax the objective when combining federated learning and adversarial training. Experimentally, we verify the rationality and effectiveness of SFAT on various benchmarked and real-world datasets with different adversarial training and federated optimization methods. The code is publicly available at: https://github.com/ZFancy/SFAT.

## 1 INTRODUCTION

Federated learning (McMahan et al., 2017) has gained increasing attention due to the concerns of data privacy and governance issues (Smith et al., 2017; Li et al., 2018; Kairouz et al., 2019; Li et al., 2020; Karimireddy et al., 2020; Khodak et al., 2021). However, training in local clients aggravates the vulnerability to adversarial attacks (Goodfellow et al., 2015; Kurakin et al., 2016; Li et al., 2021c; Sanyal et al., 2021), motivating the consideration of adversarial robustness for the federated system. For this purpose, some recent studies explore to integrate the adversarial training into federated learning (Kairouz et al., 2019; Zizzo et al., 2020; Shah et al., 2021). Federated adversarial training faces different challenges from perspectives of the distributed systems (Li et al., 2020) and the learning paradigm (Kairouz et al., 2019). Previous works mainly target overcoming the constraints in the communication budget (Shah et al., 2021) and the hardware capacity (Hong et al., 2021).

However, one critical challenge in the algorithmic aspect is that the straightforward combination of two paradigms suffers from the unexpected robustness deterioration, impeding the progress towards adversarially robust federated systems. As shown in the Figure 1(a), when considering the Federated Adversarial Training (FAT) (Zizzo et al., 2020) that directly employs adversarial training (Madry et al., 2018) in federated learning based on FedAvg (McMahan et al., 2017), one typical phenomenon is that its robust accuracy dramatically decreases at the later stage of training compared with the centralized cases (Madry et al., 2018). To the best of our knowledge, there is still a lack of in-depth understanding and algorithmic breakthroughs to overcome it, as almost all the previous explorations (Shah et al., 2021; Hong et al., 2021) still consistently adopt the conventional framework (*i.e.*, FAT).

We dive into the issue of robustness deterioration and discover that it may attribute to the intensified heterogeneity induced by adversarial training in local clients (as Figure 2 in Section 4.1). Compared

---

†Corresponding authors: Bo Han (bhanml@comp.hkbu.edu.hk) and Jiangchao Yao (Sunarker@sjtu.edu.cn).

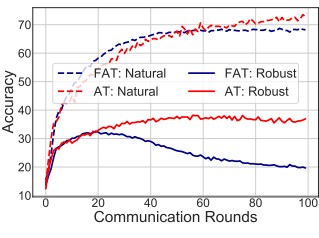 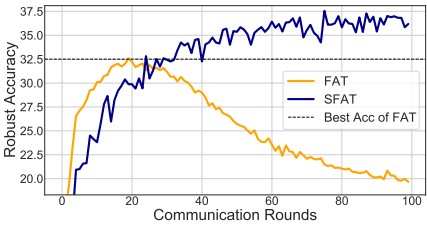

(a) Centralized AT vs. FAT          (b) FAT vs. SFAT (ours)

Figure 1: (a) comparison between centralized AT (Madry et al., 2018) and FAT (Zizzo et al., 2020) in terms of the robust accuracy and the natural accuracy. (b) comparison between FAT and SFAT (our proposed method). All the experiments of FAT and SFAT are conducted on *CIFAR-10* dataset (Non-IID) with 5 clients, and use natural test data and adversarial test data generated by PGD-20 (Madry et al., 2018) to evaluate the natural and robust accuracies. Compared with centralized AT, FAT shows performance decreasing (especially the robust accuracy) along with the learning process. In comparison, our proposed SFAT can achieve a higher robust accuracy than FAT (as indicated by the black dash line) by alleviating the deterioration. The underlying reason is elaborated out in Figure 2.

with the centralized adversarial training (Madry et al., 2018), the training data of FAT is distributed to each client, which leads to the adversarial training in each client independent from the data in the others. Therefore, the adversarial examples generated by the inner-maximization of adversarial training tend to be highly biased to each local distribution. Previous study (Li et al., 2018; 2019) indicated the local training in federated learning exhibits the optimization bias under the data heterogeneity among clients. The adversarial data generated by the biased local model even exacerbate the heterogeneity in federated optimization, making it more difficult to converge to a robust optimum.

To deal with the above challenge in the combination of adversarial training and federated learning, we propose a novel learning framework based on an $\alpha$-slack mechanism, namely, *Slack Federated Adversarial Training* (SFAT). In the high level, we relax the inner-maximization objective of adversarial training (Madry et al., 2018) into a lower bound by an $\alpha$-slack mechanism (as Eq. (1) in Section 4.2). By doing so, we construct a mediating function that asymptotically approaches the original goal while alleviating the intensified heterogeneity induced by the local adversarial generation. In detail, our SFAT assigns the client-wise slack during aggregation to upweight the clients having the small adversarial training loss (simultaneously downweight the large-loss clients), which reduces the extra exacerbated heterogeneity and alleviates the robustness deterioration (as Figure 1(b)). Theoretically, we analyze the property of our $\alpha$-slack mechanism and its benefit to achieve a better convergence (as Theorem 4.2 in Section 4.3). Empirically, we conduct extensive experiments (as Section 5 and Appendix E) to provide a comprehensive understanding of the proposed SFAT, and the results of SFAT in the context of different adversarial training and federated optimization methods demonstrate its superiority to improve the model performance. We summarize our main contributions as follows,

- We study the critical, yet thus far overlooked robustness deterioration in FAT, and discover that the reason behind this phenomenon may attribute to the intensified data heterogeneity induced by the adversarial generation in local clients (Section 4.1).

- We derive an $\alpha$-slack mechanism for adversarial training to relax the inner-maximization to a lower bound, which could asymptotically approach the original goal towards adversarial robustness and alleviate the intensified heterogeneity in federated learning (Section 4.2).

- We propose a novel framework, i.e., Slack Federated Adversarial Training (SFAT), to realize the mechanism in FAT via assigning client-wise slack during aggregation, which addresses the data heterogeneity and adversarial vulnerability in a proper manner (Section 4.3).

- We conduct extensive experiments to comprehensively understand the characteristics of the proposed SFAT (Section 5.1), as well as to verify its effectiveness on improving the model performance using several representative federated optimization methods (Section 5.2).

## 2   RELATED WORK

In this section, we briefly introduce the related work with the following aspects. More comprehensive discussions with previous literature and detailed comparison are presented in our Appendix A.

**Federated Learning.** The representative work in federated learning is FedAvg (McMahan et al., 2017), which has been proved effective during the distributed training to maintain the data privacy. To further address the heterogeneous issues, several optimization approaches have been proposed *e.g.,* FedProx (Li et al., 2018), FedNova (Wang et al., 2020b) and Scaffold (Karimireddy et al., 2020). FedProx introduced a proximal term for FedAvg to constrain the model drift cause by heterogeneity; Scaffold utilized the control variates to reduce the gradient variance in the local updates and accelerate the convergence. Mansour et al. (2022) provides an extended theoretical framework to analyze the aggregation methods and utilize the reweighting to enhance the learning stability and convergence. Note that, the intensified heterogeneity is different from the natural data heterogeneity (Li et al., 2019) that previous methods targeted, and our proposed method is orthogonal to and compatible with them.

**Adversarial Training.** As one of the defensive methods (Papernot et al., 2016), adversarial training (Madry et al., 2018) is to improve the robustness of machine learning models. The classical AT (Madry et al., 2018) is built upon on a min-max formula to optimize the worst case, *e.g.,* the adversarial example near the natural example (Goodfellow et al., 2015). Zhang et al. (2019) decomposed the prediction error for adversarial examples as the sum of the natural error and the boundary error, and proposed TRADES to balance the classification performance between the natural and adversarial examples. Wang et al. (2020c) further explored the influence of the misclassified examples on the robustness, and proposed MART that emphasizes the misclassified examples to boost the classic AT.

**Federated Adversarial Training.** Recently, several works have made the exploration on the adversarial training in the context of federated learning. To our best knowledge, Zizzo et al. (2020) takes the first trial to study the feasibility of extending federated learning (McMahan et al., 2017) with the standard AT on both IID and Non-IID settings. Considering the practical situations (Kairouz et al., 2019), the challenges of federated adversarial training are mainly from the distributed learning paradigm or the system constraints. From the learning aspect, Zizzo et al. (2020) found that there was a large performance gap existing between the federated and the centralized adversarial training, especially on the Non-IID data. From the system aspect, Shah et al. (2021) designed a dynamic schedule for local training to pursue higher robustness under the communication budget. Hong et al. (2021) explored how to effectively propagate the adversarial robustness when only limited clients in federated learning have the sufficient computational budget to afford AT. Different from above works, we are to solve the robustness deterioration issue induced by the intensified heterogeneity.

## 3 PRELIMINARIES

### 3.1 ADVERSARIAL TRAINING

Let $(\mathcal{X}, d_\infty)$ denote the input feature space $\mathcal{X}$ with the infinity distance metric $d_\infty(x, \tilde{x}) = \|x - \tilde{x}\|_\infty$, and $\mathcal{B}_\epsilon[x] = \{\tilde{x} \in \mathcal{X} \mid d_\infty(x, \tilde{x}) \leq \epsilon\}$ be the closed ball of radius $\epsilon > 0$ centered at $x$ in $\mathcal{X}$. Given a dataset $S = \{(x_n, y_n)\}_{n=1}^{N}$, where $x_n \in \mathcal{X}$ and $y_n \in \mathcal{Y} = \{0, 1, ..., C - 1\}$, the objective function of Adversarial Training (AT) (Madry et al., 2018) is a min-max formula defined as follows, $\min_{f_\theta \in \mathcal{F}} \frac{1}{N} \sum_{n=1}^{N} \max_{\tilde{x}_n \in \mathcal{B}_\epsilon[x_n]} \ell(f_\theta(\tilde{x}_n), y_n)$, where $\mathcal{F}$ is the hypothesis space, $\tilde{x}_n$ is the *most adversarial data* within the $\epsilon$-ball centered at $x_n$, $f_\theta(\cdot) : \mathcal{X} \to \mathbb{R}^C$ is a score function, $\ell : \mathbb{R}^C \times \mathcal{Y} \to \mathbb{R}$ is a composition of a base loss $\ell_B : \Delta^{C-1} \times \mathcal{Y} \to \mathbb{R}$ (*e.g.,* the Cross-Entropy loss) and an inverse link function $\ell_L : \mathbb{R}^C \to \Delta^{C-1}$ (*e.g.,* the Softmax). Here, $\Delta^{C-1}$ is the corresponding probability simplex that yields $\ell(f_\theta(\cdot), y) = \ell_B(\ell_L(f_\theta(\cdot)), y)$. For the inner-maximization, the multi-step projected gradient descent (PGD) (Madry et al., 2018) is usually employed to generate the adversarial samples. Given the natural data $x^{(0)} \in \mathcal{X}$ and a step size $\alpha > 0$, PGD works as follows, $x^{(t+1)} = \Pi_{\mathcal{B}_\epsilon[x^{(0)}]}\big(x^{(t)} + \alpha \mathrm{sign}(\nabla_{x^{(t)}} \ell(f_\theta(x^{(t)}), y))\big)$, where $t \in \mathbb{N}$, $x^{(t)}$ is the adversarial data at step $t$, $\mathrm{sign}(\cdot)$ is the function that extracts the sign of tensor elements, and $\Pi_{\mathcal{B}_\epsilon[x^{(0)}]}(\cdot)$ is the projection function that projects the adversarial data back into the $\epsilon$-ball centered at $x^{(0)}$ if necessary.

### 3.2 FEDERATED LEARNING

Let $\mathcal{D}_k$ denote a finite set of samples from the $k$-th client, and in each round, a set of datasets $\{\mathcal{D}_k\}_{k=1}^{K}$ from $K$ clients are involved into the training. The objective of federated learning is to learn a machine learning model without any exchange of the training data between the clients and the

Figure 2: Left panel: Heterogeneity measure using the client drift (Li et al., 2018; Karimireddy et al., 2020) with different adversarial generation on CIFAR-10 dataset. Right panel: comparison about FAT with our SFAT on aggregating local clients. The local adversarial generation would maximize the loss and are observed to exacerbate the heterogeneity (more empirical verification that correlates to robust deterioration can refer to Appendixes E.3 and E.4). Compared with FAT, SFAT selectively upweights/downweights the client with small/large adversarial training loss to alleviate it during aggregation, which follows our $\alpha$-slack mechanism to relax the original objective into a lower bound. We also compare the client drift of FAT and SFAT in Figures 3 and 10 to explain their difference.

server. The current popular strategy, namely FedAvg, is introduced by McMahan et al. (2017), where the clients collaboratively send the locally trained model parameters $\theta_k$ to the server for the global average aggregation. Concretely, each client runs on a local copy of the global model (parameterized by $\theta^t$ in the $t$-th round) with its local data to optimize the learning objective. Then, the server receives their updated model parameters $\{\theta_k^t\}_{k=1}^K$ of all clients and performs the following aggregation, $\theta^{t+1} = \frac{1}{N} \sum_{k=1}^K N_k \theta_k^t$, where $N_k$ denotes the number of samples in $\mathcal{D}_k$ and $N = \sum_{k=1}^K N_k$. Then, the parameters $\theta^{t+1}$ for the global model will be sent back to each client for another round of training. After sufficient rounds of such a periodic aggregation, we expect the stationary point of federated learning will approximately approach to or have a small gap with that from the centralized case.

## 4 SLACK FEDERATED ADVERSARIAL TRAINING

In this section, we first discuss the motivation of the problem when directly applying adversarial training into federated learning. Then, we propose $\alpha$-slack mechanism and show some theoretical insights. Finally, we propose our Slack Federated Adversarial Training (SFAT) to realize the $\alpha$-slack mechanism and provide its corresponding convergence analysis and insights.

### 4.1 MOTIVATION

Considering federated learning, one of the primary difficulties is the biased optimization caused by the local training with heterogeneous data (Zhao et al., 2018; Li et al., 2018; 2020). As for adversarial training, the key distinction from standard training is the use of inner-maximization to generate adversarial data, which pursues the better adversarial robustness. When combining the two learning paradigms, we conjecture that the following issue may arise especially under the Non-IID case,

> *the inner-maximization for pursuing adversarial robustness would exacerbate the data heterogeneity among local clients in federated learning.*

Intuitively, the adversarial data inherits the optimization bias since its generation is on the basis of the biased local model. As the adversarial strength increases, i.e., when adversarial training trying to gain more robustness via stronger adversarial generation, the heterogeneity are observed to be exacerbated (indicated by the client drift in the left of Figure 2). In the right of Figure 2, we illustrate the aggregation procedure with two clients (i.e., Clients A and B). With adversarial generation on local data, the optimization bias of each client is exacerbated, which may result in more severe heterogeneity and the poor convergence. Therefore, a new mechanism is required to prevent the intensified heterogeneity that shows detrimental to convergence in FAT (refer to Figures 9 and 10).

### 4.2 $\alpha$-SLACK MECHANISM

As previous analysis, the inner-maximization of adversarial training is incompatible with federated learning due to the intensification of data heterogeneity. To deal with that, one possible way is to build a mediating function that can alleviate the intensified heterogeneity effect and simultaneously approach the goal of the original objective asymptotically to pursue adversarial robustness. To

this intuition, we consider a slack of the inner-maximization to prevent the intensification of the optimization bias as illustrated in the right panel of Figure 2. Formally, we decompose the inner-maximization in adversarial training into the independent $K$ populations that correspond to the $K$ clients in federated learning, and relax it into a lower bound by $\alpha$ as follows[1],

$$
\begin{aligned}
\mathcal{L}_{AT} &= \frac{1}{N} \sum_{n=1}^{N} \max_{\tilde{x}_n \in \mathcal{B}_\epsilon[x_n]} \ell(f_\theta(\tilde{x}_n), y_n) = \sum_{k=1}^{K} \frac{N_k}{N} \underbrace{\left( \frac{1}{N_k} \sum_{n=1}^{N_k} \max_{\tilde{x}_n \in \mathcal{B}_\epsilon[x_n]} \ell(f_\theta(\tilde{x}_n^k), y_n^k) \right)}_{\mathcal{L}_k} \\
&\geq (1+\alpha) \sum_{k=1}^{\widehat{K}} \frac{N_{\phi(k)}}{N} \mathcal{L}_{\phi(k)} + (1-\alpha) \sum_{k=\widehat{K}+1}^{K} \frac{N_{\phi(k)}}{N} \mathcal{L}_{\phi(k)} \\
&\doteq \mathcal{L}^\alpha(\widehat{K}), \qquad \text{s.t. } \alpha \in [0,\ 1),\ \widehat{K} \leq \frac{K}{2},
\end{aligned}
\tag{1}
$$

where $\phi(\cdot)$ is a function which maps the index to the original population sorted by $\{\frac{N_k}{N}\mathcal{L}_k\}$ in an ascending order. Note that, $\phi(\cdot)$ is an auxiliary operation that is non-parametric and does not affect the gradient back-propagation. The intuition is to relax the original objective to alleviate the heterogeneity exacerbation, like the classical continuous relaxation (e.g., Gumbel-SoftMax (Jang et al., 2017)) in the discrete optimization. We have also discuss the reverse operation of $\alpha$-slack in experiments (refer to Section 5.1 and Appendix E.3), which pursues an upper bound but it further exacerbates the heterogeneity. The following Theorems 4.1 and 4.2 will provide more analysis about the characteristics of this slack mechanism and their complete proofs are given in Appendix B.

**Theorem 4.1.** $\mathcal{L}^\alpha(\widehat{K})$ *is monotonically decreasing w.r.t. both $\alpha$ and $\widehat{K}$, i.e., $\mathcal{L}^{\alpha_1}(\widehat{K}) < \mathcal{L}^{\alpha_2}(\widehat{K})$ if $\alpha_1 > \alpha_2$ and $\mathcal{L}^\alpha(\widehat{K}_1) < \mathcal{L}^\alpha(\widehat{K}_2)$ if $\widehat{K}_1 > \widehat{K}_2$. Specifically, $\mathcal{L}^\alpha(\widehat{K})$ recovers $\mathcal{L}$ of adversarial training when $\alpha$ achieves 0, and $\mathcal{L}^\alpha(\widehat{K})$ relaxes $\mathcal{L}$ to a lower bound objective by increasing $\widehat{K}$ and $\alpha$.*

Based on the above theorem, we can flexibly emphasize the importance of partial populations by setting the proper hyperparameters (i.e., $\widehat{K}$ and $\alpha$), to alleviate the evenly averaging of the harsh heterogeneous updates in FAT as illustrated by the right-most panel of Figure 2.

**Theorem 4.2.** *Assume the loss function $\ell(\cdot, \cdot)$ in Eq. (1) satisfies the Lipschitzian smoothness condition w.r.t. the model parameter $\theta$ and the training sample $x$, and is $\lambda$-strongly concave for all $x$, and $\mathbb{E}\left[ ||\nabla_\theta \mathcal{L}^\alpha(\widehat{K}) - \nabla_\theta \ell(f_\theta(\tilde{x}), y)||_2^2 \right] \leq \delta^2$, where $\tilde{x}$ is the adversarial example. Then, after the sufficient $T$-step optimization i.e., $T \geq \frac{L\Delta}{\delta^2}$, for the $\alpha$-slack mechanism of decomposed Adversarial Training with the constant stepsize $\sqrt{\frac{\Delta}{LT\sigma^2}}$ in PGD, we have the following convergence property,*

$$
\frac{1}{T} \sum_{t=1}^{T} \mathbb{E}\left[ \left\| \nabla \mathcal{L}^\alpha(\widehat{K})|_{\theta^t} \right\|_2^2 \right] \leq \left( 1 + \alpha \frac{\frac{1}{T}\sum_{t=1}^{T} \xi^{(t)}}{N} \right) \left( \frac{4L_{\theta x}^2 \epsilon}{\lambda} + 4\delta \sqrt{\frac{L\Delta}{T}} \right),
\tag{2}
$$

*where $L = L_{\theta\theta} + \frac{L_{\theta x} L_{x\theta}}{\lambda}$ defined by the Lipschitzian constraints, $\Delta \geq \mathcal{L}^\alpha(\widehat{K})|_{\theta^0} - \inf_\theta \mathcal{L}^\alpha(\widehat{K})$ and $\xi^{(t)} = \sum_{k=1}^{\widehat{K}} N_{\phi(k)}^{(t)} - \sum_{\widehat{K}+1}^{K} N_{\phi(k)}^{(t)}$ meaning the accumulative counting difference of the $t$-th step.*

The complete notation explanation can be found in Appendix B.2. Note that, when $\alpha = 0$, Eq. (1) recovers the original loss of Adversarial Training, and the first part in the RHS of Eq. (2) goes to 1 that recovers the convergence rate of Adversarial Training (Sinha et al., 2018). When $\alpha \to 1$, Eq. (1) becomes more biased, while simultaneously the straightforward benefit is that we can achieve a faster convergence in Eq. (2) if $\frac{1}{T}\sum_{t=1}^{T} \xi^{(t)} < 0$, i.e., $\left( 1 + \alpha \frac{\frac{1}{T}\sum_{t=1}^{T} \xi^{(t)}}{N} \right) < 1$. Actually, this is possible when the sample number is approximately similar among all clients and the top-1 choice easily has $-N < \xi^{(t)} = N_{\phi(1)}^{(t)} - \sum_{k=2}^{K} N_{\phi(k)}^{(t)} < 0$ in each optimization step. In this case, a larger $\alpha$ has a faster convergence. Therefore, the proposed $\alpha$-slack mechanism of adversarial training provides us a way to acquire the optimization benefits in FAT by introducing the small relaxation to the objective.

---

[1]Note that the intuition here is to find a mechanism to *relax* the original objective like the classical continuous relaxation in the discrete optimization. Pursuing an upper bound does not help but exacerbates the heterogeneity.

### 4.3 REALIZATION OF SLACK FEDERATED ADVERSARIAL TRAINING

Based on the previous analysis of the $\alpha$-slack mechanism, we propose a *Slack Federated Adversarial Training* to combine adversarial training and federated learning. The intuition is applying the $\alpha$-slack mechanism into the inner-maximization in FAT, which can be formalized as follows,

$$\min \mathcal{L}_{\text{SFAT}} = \min_{f_\theta \in \mathcal{F}} \frac{1}{\sum_k^K N_k} \sum_{k=1}^K P_k N_k \cdot \underbrace{\left( \frac{1}{N_k} \sum_{n=1}^{N_k} \max_{\tilde{x}_n^k \in \mathcal{B}_\epsilon[x_n^k]} \ell(f_\theta(\tilde{x}_n^k), y_n^k) \right)}_{\mathcal{L}_k}, \tag{3}$$

where $P_k$ denotes the weight assigned to the $k$-th client based on the ascending sort of weighted client losses compared with the $\widehat{K}$-th one, which can be $(1+\alpha)\cdot\mathbb{1}\left(\frac{N_k}{N}\mathcal{L}_k \leq \mathcal{L}_{\text{sorted}}[\widehat{K}]\right)+(1-\alpha)\cdot\mathbb{1}\left(\frac{N_k}{N}\mathcal{L}_k > \mathcal{L}_{\text{sorted}}[\widehat{K}]\right)$ that corresponds to the relaxed loss in Eq. (1). For simplicity without loss of generality, we can transform the slack mechanism as the weight adjustment of those selected clients with smaller adversarial training losses, and assign $P_k = ((1+\alpha)/(1-\alpha)\cdot\mathbb{1}\left(\frac{N_k}{N}\mathcal{L}_k \leq \mathcal{L}_{\text{sorted}}[\widehat{K}]\right)+1\cdot\mathbb{1}\left(\frac{N_k}{N}\mathcal{L}_k > \mathcal{L}_{\text{sorted}}[\widehat{K}]\right))/((\sum_{k=1}^K P_k) + 2\alpha/(1-\alpha))$ to ensure the lower bound derivation. For more details, we summarize the procedure of SFAT in Algorithm 1 of Appendix D, which consists of multi-round iterations between the local training on the client side and the global aggregation on the server side.

Concretely, on the client side, after downloading the global model parameter from the server, each client will perform the adversarial training on its local data. At the same time, the client loss on the adversarial examples is also recorded, which acts as the soft-indicator of the local bias induced by the radical adversarial generation. Then, when the training steps reach to the condition, the client will upload its model parameter and the loss to the server. On the server side, after collecting the model parameters $\{\theta_k\}_{k=1}^K$ and the losses $\{\mathcal{L}_k\}_{k=1}^K$ of all clients, it will first sort $\{\frac{N_k}{N}\mathcal{L}_k\}_{k=1}^K$ in an ascending order to find the top-$\widehat{K}$ clients. Based on that, the global model parameters will be aggregated by the $\alpha$-slack mechanism in which the model parameters of the top-$\widehat{K}$ clients are upweighted with $(1 + \alpha)/(1 - \alpha)$ and the remaining is downweighted. For atypical layers (Li et al., 2021b) *e.g.,* BN, it is outside the scope of this paper and we keep the aggregation same as FedAvg.

In Figure 3, we empirically justify the rationality of our SFAT as illustrated by Figure 2. We employ the averaged client drift (Li et al., 2018; Karimireddy et al., 2020), i.e., $\|\theta_k - \theta_s\|_2$ (the parameter difference between local model and averaged global model), to approximately reflect the effect of data heterogeneity (illustrated as the diverse optimization directions in the right panel of Figure 2). From Figure 3, we can see the client drift of SFAT is smaller than that of FAT in the later stages. This indicates the optimization directions of clients are less diverse, contributing to the alleviation of the intensified heterogeneity. In Figure 7, we trace the index of the top-weighted client in one experiment and find it dynamically routes among different clients instead of a fixed one.

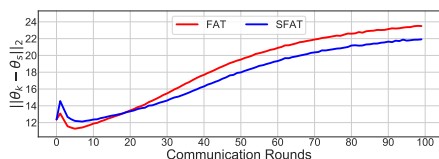

Figure 3: The averaged client drift (Li et al., 2018; Karimireddy et al., 2020) in each communication round on *CIFAR-10* (Non-IID). SFAT generally achieves a smaller drift compared to FAT, *i.e.,* a less heterogeneous aggregation. More empirical results can refer to Appendixes E.3, E.4 and E.8.

In the following, we provide the theoretical analysis of our SFAT on the convergence in the context of federated learning (Li et al., 2019), which is slightly different from the centralized counterpart.

**Theorem 4.3.** *Assume the loss function $\ell(\cdot, \cdot)$ in Eq. (3) is $L$-smooth and $\lambda$-strongly convex w.r.t. the model parameter $\theta$, and the expected norm and the variance of the stochastic gradient in each client respectively satisfy $\mathbb{E}\left[\|\nabla_\theta\ell(f_\theta(\tilde{x}^k), y^k)\|_2^2\right] \leq \varsigma^2$ and $\mathbb{E}\left[\|\nabla_\theta\ell(f_\theta(\tilde{x}^k), y^k) - \nabla_\theta\mathcal{L}_k\|_2^2\right] \leq \delta_k^2$. Let $\kappa = \frac{L}{\lambda}$, $\gamma = \max\{8\kappa, E\}$ where $E$ is the iteration number of the local adversarial training with the learning rate $\eta_t = \frac{2}{\lambda(\gamma+t)}$. Then, after the sufficient $T$-step communication rounds for SFAT, we have the following asymptotics to the optimal,*

$$\mathbb{E}[\mathcal{L}_{SFAT}] - \mathcal{L}^* \leq \frac{\kappa}{\gamma + T - 1}\left(\frac{2B}{\lambda} + \frac{\lambda\gamma}{2}\mathbb{E}\left[\|\theta^0 - \theta^*\|^2\right]\right), \tag{4}$$

*where $\mathcal{L}^*$ is the minimum value of $\mathcal{L}_{SFAT}$, $\theta^*$ is the optimal model parameter, and*

$$B = \sum_{k=1}^{K} \left( \frac{P_k^{(T)}}{1 + \alpha \frac{\xi^{(T)}}{N}} \right)^2 \left( \frac{N_k}{N} \delta_k \right)^2 + 6L \left( \mathcal{L}^* - \sum_{k}^{K} \frac{P_k^{(T)}}{1 + \alpha \frac{\xi^{(T)}}{N}} \frac{N_k}{N} \mathcal{L}_k^* \right) + 8(E-1)^2 \varsigma^2.$$

When $\alpha = 0$, we have $P_k^{(T)}/(1 + \alpha \frac{\xi^{(T)}}{N}) = 1$ and Eq. (4) becomes the convergence rate of FedAvg on Non-IID data (Li et al., 2019). Different from Theorem 4.2 that concludes in the centralized training setting, when $\alpha \to 1$, the convergence is indefinite compared to the standard FAT, since the emerging terms in $B$, *i.e.,* $\left( P_k^{(T)}/(1 + \alpha \frac{\xi^{(T)}}{N}) \right)^2$ and $P_k^{(T)}/(1 + \alpha \frac{\xi^{(T)}}{N})$, are acted as the scalar timing by the personalized variance bound $\delta_k^2$ and the local optimum $\mathcal{L}_k^*$ of each client. One possible case is when the optimization approaches to the optimal parameter $\theta^*$, $\delta_k^2$ can be in a smaller scale relative to the scale of $\mathcal{L}_k^*$. In this case, the increment of the first term of $B$ can be totally counteracted by the loss of the second term of $B$ so that in sum $B$ becomes smaller. Then, we can have a tighter upper bound for SFAT in Eq. (4) to achieve a faster convergence than FAT. The completed proof of Theorem 4.3 is given in Appendix B.3 and the empirical verification is presented in Appendix E.7. The following section will comprehensively confirm that SFAT can reach to a more robust optimum. Note that, the theorem we deduce here follows the same assumptions (Li et al., 2019) and is mainly to compare the difference with the early convergence analysis in federated learning. However, some relaxation on the assumptions could be further improved by considering more practical SGD theories (Nguyen et al., 2018), which is beyond the scope of this paper and we leave in the future works.

## 5 EXPERIMENTS

In this section, we provide a comprehensive analysis of SFAT and empirically verify its effectiveness compared with the current methods on a range of benchmarked datasets and a real-world dataset. The source code is publicly available at: `https://github.com/ZFancy/SFAT`.

**Setups.** We conduct the experiments on three benchmark datasets, *i.e., CIFAR-10*, *CIFAR-100* (Krizhevsky, 2009), *SVHN* (Netzer et al., 2011) as well as a real-world dataset *CelebA* (Caldas et al., 2018) for federated adversarial training. For the IID scenario, we just randomly and evenly distribute the samples to each client. For the Non-IID scenario, we follow McMahan et al. (2017); Shah et al. (2021) to partition the training data based on their labels. To be specific, a skew parameter $s$ is utilized in the data partition introduced by Shah et al. (2021), which enables $K$ clients to get a majority of the data samples from a subset of classes. We denote the set of all classes in a dataset as $\mathcal{Y}$ and create $\mathcal{Y}_k$ by dividing all the class labels equally among $K$ clients. Accordingly, we split the data across $K$ clients that each client has $(100 - (K-1) \times s)\%$ of data for the class in $\mathcal{Y}_k$ and $s\%$ of data in other split sets. In the test phase, we evaluate the model's standard performance using natural test data and its robust performance using adversarial test data generated by FGSM (Goodfellow et al., 2015), PGD-20, $CW_\infty$ (Carlini & Wagner, 2017) and AutoAttack (Croce & Hein, 2020) (termed as AA) to evaluate its robust performance. More detailed settings are provided in Appendix E.

### 5.1 ABLATION STUDY

In this subsection, we conduct various experiments on *CIFAR-10* with the Non-IID setting to visualize the characteristics of our proposed SFAT. More comprehensive results are provided in Appendix E.

**Non-AT vs. AT.** In the left two panels of Figure 4, we respectively apply our $\alpha$-slack mechanism to Federated Standard Training (SFST) and Federated Adversarial Training (SFAT) under $\widehat{K} = 1$. We also consider both FedAvg and FedProx in this experiment to guarantee the universality. From the curves, we can see that SFST has the negative effect on the natural accuracy, while SFAT consistently improves the robust accuracy based on FedAvg and FedProx. This indicates our $\alpha$-slack mechanism is tailored for the inner-maximization of Federated Adversarial Training instead of the outer-minimization considered by other federated optimization methods. In addition, we also verify the orthogonal effects of SFAT with FedProx on reducing client drift in Appendix E.8 and strengthen the hyper-parameter of FedProx to demonstrate the consistent effectiveness of our SFAT.

**Re-SFAT vs. SFAT.** In the middle panel of Figure 4, we conversely apply our $\alpha$-slack mechanism in Federated Adversarial Training (Re-SFAT) where we upweight the client with large adversarial

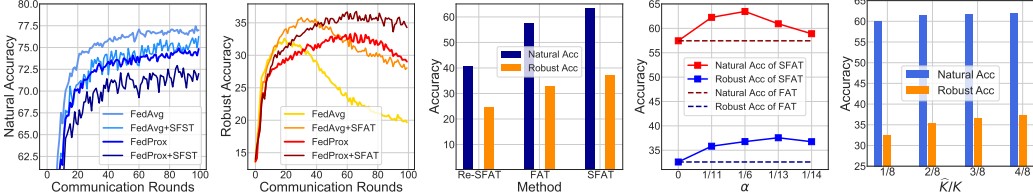

Figure 4: Ablation study on SFAT. Left two panels: comparison between federated standard training and federated adversarial training respectively in combination with the $\alpha$-slack mechanism, *i.e.,* SFST vs. SFAT ($\alpha = 1/11$). Middle panel: comparison between FAT, SFAT and Re-SFAT (the reverse operation which upweights the clients with large adversarial training loss). Right two panels: the natural accuracy and the robust accuracy of SFAT with different $\alpha$ and different $\widehat{K}$ on CIFAR-10.

Table 1: Test accuracy on *CIFAR-10* (Non-IID) partition with different client numbers.

| Client Number | Methods | Natural | PGD-20 | CW$_\infty$ |
|---|---|---|---|---|
| 10 | FAT | 56.62% | 31.24% | 29.82% |
| | **SFAT** | **56.67%** | **33.31%** | **31.58%** |
| 20 | FAT | 60.55% | 32.67% | 31.07% |
| | **SFAT** | **62.24%** | **35.66%** | **33.21%** |
| 25 | FAT | 58.97% | 32.98% | 31.14% |
| | **SFAT** | **62.73%** | **35.75%** | **33.16%** |
| 50 | FAT | 56.74% | 32.91% | 30.50% |
| | **SFAT** | **57.21%** | **34.35%** | **31.75%** |

Table 2: Test accuracy on *CIFAR-10* (Non-IID) with different local adversarial training methods.

| Methods | | Natural | PGD-20 | CW$_\infty$ |
|---|---|---|---|---|
| AT | FAT | 57.45% | 32.58% | 30.52% |
| | **SFAT** | **62.34%** | **35.59%** | **33.06%** |
| TRADES | FAT | 64.00% | 31.64% | 28.95% |
| | **SFAT** | **65.26%** | **35.10%** | **31.80%** |
| MART | FAT | 56.29% | 36.27% | 32.41% |
| | **SFAT** | **58.41%** | **38.90%** | **34.67%** |

training loss. Note that the Re-SFAT share a similar spirit with the Agnostic Federated Learning (AFL) (Mohri et al., 2019), which seeks to improve the generalization in standard federated learning through a loss-maximization reweighting. Through its comparison with FAT and SFAT, we can see enlarging the inner-maximization during aggregation could severely exacerbate heterogeneity which results in the worse performance, while our SFAT improves the performance through relaxing it into a lower bound. We also confirm the rationality of SFAT on other datasets in Appendix E.3.

**Impact of $\alpha$ and $\widehat{K}$.** To study the effect of hyperparameters in SFAT, we conduct several ablation experiments to verify the model performance. Regarding the experiments of $\alpha$, we set the client number $K = 5$ and $\widehat{K} = 1$ to upweight/downweight the client models in each communication round. The right middle panel of Figure 4 shows that $\alpha \in (0, 1/6]$ can significantly improve the robust accuracy and the natural accuracy, while a larger $\alpha$ might be inappropriate to the natural accuracy. We also conduct a comprehensive ablation to show the effects of $\alpha$ during training in Appendix E.4. Regarding the choice of $\widehat{K}$, we specially set $K = 8$ in this experiment to span the range of $\widehat{K}$ due to the constraint $\widehat{K} <= K/2$. The right panel of Figure 4 tracks the accuracy of SFAT with increasing $\widehat{K}$. As can be seen, both natural accuracy and robust accuracy are improved even with larger $\widehat{K}$, which shows the effect of $\widehat{K}$ on the slack of inner-maximization. For other basic experimental setting, e.g., the local epochs, we explore the effectiveness under different setups in Appendix E.2.

**Different client numbers.** In Table 1, we validate our SFAT on training with different client numbers, where we set $\alpha = 1/11$ (i.e., $\frac{1+\alpha}{1-\alpha} = 1.2$) and $\widehat{K} = K/5$. The results show that with the client number varying from 10 to 50, our SFAT can consistently gain better natural and robust performance than FAT. We also confirm the scalability of using other datasets (in Appendix E.9), the performance on unequal data splits in the different clients (in Appendix E.10), as well as the effectiveness in a practical situation where only a subset of clients participate in the aggregation (in Appendix E.11).

**Different adversarial training methods.** In Table 2, we validate the combination of $\alpha$-slack mechanism and different adversarial training methods (*i.e.,* AT (Madry et al., 2018), TRADES (Zhang et al., 2019) and MART (Wang et al., 2020c)), where we switch different local adversarial training methods on the client side. Through the comparison with FAT, the results show that SFAT can consistently boost both the natural performance and the robust performance, and is applicable to other state-of-the-art adversarial training methods under the federated learning scenarios.

Table 3: Performance on Non-IID settings with different federated optimization methods (Mean±Std).

| Setting | | Non-IID | | | | |
|---|---|---|---|---|---|---|
| CIFAR-10 | | Natural | FGSM | PGD-20 | CW$_\infty$ | AA |
| FedAvg | FAT | 58.13±0.68% | 40.06±0.62% | 32.56±0.01% | 30.88±0.37% | 29.17±0.03% |
| | **SFAT** | **63.36±0.07%** | **44.82±0.32%** | **37.14±0.03%** | **33.39±0.61%** | **31.66±0.70%** |
| FedProx | FAT | 59.95±0.45% | 41.44±0.15% | 33.83±0.01% | 31.65±0.36% | 30.11±0.09% |
| | **SFAT** | **62.04±0.47%** | **44.21±0.08%** | **36.64±0.11%** | **32.62±0.20%** | **31.83±0.15%** |
| Scaffold | FAT | 61.44±1.37% | 42.85±0.76% | 34.08±0.05% | 32.56±0.02% | 31.03±0.08% |
| | **SFAT** | **63.16±0.96%** | **45.55±0.50%** | **37.33±0.02%** | **34.82±0.04%** | **33.32±0.01%** |
| CIFAR-100 | | Natural | FGSM | PGD-20 | CW$_\infty$ | AA |
| FedAvg | FAT | 34.63±0.56% | 19.92±0.28% | 15.40±0.20% | 13.23±0.03% | 12.23±0.01% |
| | **SFAT** | **35.65±0.54%** | **20.23±0.44%** | **16.24±0.16%** | **13.53±0.02%** | **12.45±0.03%** |
| FedProx | FAT | 31.93±0.43% | 19.06±0.17% | 15.30±0.08% | 12.93±0.02% | 12.01±0.04% |
| | **SFAT** | **34.87±0.24%** | **20.54±0.08%** | **16.09±0.10%** | **13.35±0.12%** | **12.44±0.20%** |
| Scaffold | FAT | 39.98±0.02% | 24.30±0.04% | 19.34±0.07% | 16.49±0.12% | 15.29±0.08% |
| | **SFAT** | **44.13±0.05%** | **25.32±0.94%** | **20.22±0.07%** | **16.96±0.17%** | **15.80±0.10%** |
| SVHN | | Natural | FGSM | PGD-20 | CW$_\infty$ | AA |
| FedAvg | FAT | **91.52±0.28%** | 88.13±0.18% | 68.98±0.11% | 68.04±0.15% | 66.59±0.04% |
| | **SFAT** | 91.26±0.01% | **88.27±0.02%** | **72.04±0.32%** | **69.96±0.16%** | **68.89±0.27%** |
| FedProx | FAT | 91.00±0.08% | 87.65±0.15% | 68.48±0.04% | 67.16±0.02% | 65.76±0.18% |
| | **SFAT** | **91.19±0.06%** | **88.15±0.01%** | **71.84±0.30%** | **69.88±0.35%** | **68.84±0.37%** |
| Scaffold | FAT | 90.82±0.87% | 87.89±0.66% | 69.51±0.84% | 68.12±0.88% | 67.19±0.54% |
| | **SFAT** | **90.93±0.76%** | **88.27±0.45%** | **71.77±0.38%** | 69.49±0.67% | **68.37±0.48%** |

## 5.2 PERFORMANCE EVALUATION

Next, we compare SFAT with FAT on various benchmark datasets to verify its effectiveness. Specifically, we validate it with three representative federated optimization methods, *i.e.,* FedAvg, FedProx and Scalffold. Considering the sensitivity of data selection in Non-IID setting, we report the results with Mean ± Std values in Table 3 after running multiple times. For the completeness of experiments, we also demonstrate the efficacy of SFAT on the IID setting and a real-world dataset. The overall results for comparison (with the centralized counterparts) are presented in Appendixs E.11 and E.12.

According to Table 3 on *CIFAR-10*, we can find that our SFAT significantly outperforms FAT on the Non-IID data in terms of both the natural accuracy (∼2%-6%) and the robust accuracy (∼2%-5%). As for FedProx and Scaffold which are specifically designed to handle the heterogeneous issues in federated learning, employing them in FAT can indeed improve the model performance compared with that based on FedAvg. Our SFAT further boosts the performance by alleviating the extra heterogeneity from adversarial training. On *CIFAR-100* and *SVHN*, we can find the similar improvement in Table 3 as that of *CIFAR-10* under three types of federated optimization methods.

## 6 CONCLUSION

In this work, we investigated the issue of robustness deterioration when combining adversarial training with federated learning, and revealed that it may attribute to the intensified heterogeneity induced by local adversarial generation. To alleviate it, we introduce an $\alpha$-slack decomposed mechanism into adversarial training to relax the overall inner-maximization. Based on this, we propose a new framework, i.e., Slack Federated Adversarial Training (SFAT). We provide both the theoretical analysis and empirical evidences to understand the proposed method. The experimental results under various settings confirm the consistent effectiveness of our proposed SFAT. Nevertheless, we only move a small step on the intensified heterogeneous issue in the combination of two learning paradigms, federated adversarial training still suffers from the other challenges of systems or algorithms. Beyond the empirical conjecture in the problem focused on this work, more theoretical understanding on the dynamical heterogeneous issue under federated learning is worthwhile to explore in the future.

## ACKNOWLEDGEMENT

JNZ and BH were supported by NSFC Young Scientists Fund No. 62006202, Guangdong Basic and Applied Basic Research Foundation No. 2022A1515011652, RGC Early Career Scheme No. 22200720, RGC Research Matching Grant Scheme No. RMGS20221102, No. RMGS20221306 and No. RMGS20221309. BH was also supported by CAAI-Huawei MindSpore Open Fund and HKBU CSD Departmental Incentive Grant. TLL was partially supported by Australian Research Council Projects IC-190100031, LP-220100527, DP-220102121, and FT-220100318. JLX was partially supported by Hong Kong RGC Grants 12202221 and C2004-21GF.

## ETHICS STATEMENT

This paper does not raise any ethics concerns. This study does not involve any human subjects, practices to data set releases, potentially harmful insights, methodologies and applications, potential conflicts of interest and sponsorship, discrimination/bias/fairness concerns, privacy and security issues, legal compliance, and research integrity issues.

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

APPENDIX

The Appendix is organized as follows. In Appendix A, we detailedly discuss the related work and highlighted our distinguishable point compared with previous work. In Appendix B, we formally prove the aforementioned Equations and Theorems. In Appendix C, we demonstrate the issue of bias exacerbation (as illustrated in Figure 2) using a binary classification experiment. In Appendix D, we provide illustration of the learning framework for our SFAT. In Appendix E, we present our experimental details and more quantitative results about understanding our proposed SFAT.

## A  DETAILED DISCUSSION AND COMPARISON ABOUT RELATED WORK

In this section, we detailedly discuss the related work in federated learning, adversarial training as well as the federated adversarial training. At the end of each part, we also highlight our distinguishable points compared with the previous work, either from conceptual or technical perspectives.

**Federated Learning.**    The representative work in federated learning is FedAvg (McMahan et al., 2017), which has been proved effective during the distributed training to maintain the data privacy. To further address the heterogeneous issues, several optimization approaches have been proposed *e.g.,* FedProx (Li et al., 2018), FedNova (Wang et al., 2020b) and Scaffold (Karimireddy et al., 2020). FedProx introduced a proximal term for FedAvg to constrain the model drift cause by heterogeneity; FedNova proposed a general framework that eliminated the objective inconsistency and preserved the fast convergence; Scaffold utilized the control variates to reduce the gradient variance in the local updates and accelerate the convergence. MOON (Li et al., 2021a) alleviated the heterogeneity by maximizing the agreement between the representations of the local and global models, which helps the local training of individual parties. Reisizadeh et al. (2020) developed a robust federated learning algorithm to against distribution shifts in client samples. Mohri et al. (2019) proposed Agnostic Federated Learning (AFL) to improve the fairness and generalization to different data distribution through a loss-maximization reweighting. However, it is not suitable to the problem of FAT and actually opposite to our relaxation. We show AFL performs even worse than FAT in Appendix E.3. Our SFAT introduces the $\alpha$-slack mechanism to combat intensified heterogeneity in federated adversarial training, which is orthogonal to and compatible with the most previous methods.

One concurrent work (Mansour et al., 2022) recently provide an extended theoretical framework to analyze the general aggregation methods in federated learning, it also utilizes the reweighting mechanism and propose the FedSoftBetter. However, our proposed SFAT and FedSoftBetter have both different motivations and underlying principles. On the one hand, we introduce $\alpha$-slack mechanism to alleviate the exacerbated heterogeneity, while Mansour et al. (2022) employs reweighting to enhance the stability and convergence of original FedAvg. On the other hand, SFAT focuses on federated adversarial training and origins from our $\alpha$-slack mechanism, while FedSoftBetter focuses on ordinary federated learning and originates as a specific strategy for federated aggregation from the theoretical analysis on convergence bound. Different from FedSoftBetter, as empirically verified in "Non-AT v.s. AT" of Section 5.1, our SFAT is not for ordinary federated learning but tailored for federated adversarial training. In practice, the technical forms of two weighting mechanisms are also different. SFAT directly ranks the losses of all clients and constructs the weights, while the weights of FedSoftbetter build upon the gap between the client loss and the client optimal loss. We also provide an empirical comparison of the two methods in our Appendix E.5.

**Adversarial Training.**    As one of the defensive methods (Papernot et al., 2016), adversarial training (Madry et al., 2018; Zhang et al., 2019; Jiang et al., 2020; Chen et al., 2021; Zhang et al., 2021) is to improve the robustness of machine learning models. The classical AT (Madry et al., 2018) is built upon on a min-max formula to optimize the worst case, *e.g.,* the adversarial example near the natural example (Goodfellow et al., 2015). Zhang et al. (2019) decomposed the prediction error for adversarial examples as the sum of the natural error and the boundary error, and proposed TRADES to balance the classification performance between the natural and adversarial examples. Wang et al. (2020c) further explored the influence of the misclassified examples on the robustness, and proposed MART that emphasizes the minimization of the misclassified examples to boost the AT. Zhang et al. (2020b); Sanyal et al. (2021) investigated the instance-level difficulties in AT and designed different training strategies to improve the model performance. Different from their instance-level granular-

ity, our SFAT framework leverages the client-level measure to alleviate the heterogeneous issue in the straightforward combination of adversarial training and federated learning. It is compatible to incorporate those centralized adversarial training methods (as shown in Table 2 of Section 5.1) in the updates of local clients to further improve the model performance or address some special issues.

**Federated Adversarial Training.**   Recently, several works have made the exploration on the adversarial training in the context of federated learning, which consider the data privacy and the robustness in one framework. To our best knowledge, Zizzo et al. (2020) takes the first trial to study the feasibility of extending federated learning (McMahan et al., 2017) with the standard AT on both IID and Non-IID settings. Empirically, they found that there was a large performance gap existing between the distributed and the centralized adversarial training, especially on the Non-IID data. Shah et al. (2021) designed a dynamic schedule for the local training to pursue a larger robustness under the constrained communication budget of federated learning. Hong et al. (2021) explored how to effectively propagate the adversarial robustness when only limited clients in federated learning have the sufficient computational budget to afford AT. Zhang et al. (2020a); Zizzo et al. (2021) studied to defend the malicious clients that poison the global model in federated learning, which also discuss the robustness but actually another research topic compared with the federated adversarial training in this work. Different from previous works (in Table 4), we explore to solve the robustness deterioration issue (as shown in Figure 1(a)) induced by the intensified heterogeneity (as illustrated in Figure 2) of the direct combination of federated learning and adversarial training.

Table 4: Brief comparison with some related work in federated adversarial training

| Research work | adopt FAT | Other research topic | Notable differences with ours |
|---|---|---|---|
| Shah et al. (2021) | ✓ | | focus on limited constrained communication budget |
| Hong et al. (2021) | ✓ | | assumes some client can not perform AT locally |
| Zhang et al. (2020a) | | ✓ | focus on poisoning defense |
| Zizzo et al. (2021) | | ✓ | focus on poisoning defense |

Considering the practical requirement in the real-world applications (Kairouz et al., 2019), federated adversarial training still faces various challenges. In general, these challenges mainly come from perspectives of the distributed systems (Li et al., 2020) and the unique learning paradigm (Kairouz et al., 2019). From the perspective of distributed systems, the major issue for federated adversarial training is its characteristic of high computational-cost (Madry et al., 2018). It is well known in conventional adversarial training as the local adversarial generation always require multiple times of optimization to better optimize the inner-maximization, which is to pursue the better empirical adversarial robustness (Goodfellow et al., 2015). In federated setting, we also need to consider the heterogeneous devices (Kairouz et al., 2019) in practical situation like the previous work (Hong et al., 2021) focused on. The clients with low computational capacity not only affect the synchronous aggregation but also may not affordable for the local adversarial training. Such issues in hardware also results in severe problem for the distributed learning. From the perspective of the learning paradigm, the issues is more about the training data and inference threaten for federated adversarial training. Except for the intensified heterogeneity discussed in our work, the heterogeneity data itself is also a severe problem for federated adversarial training. More conventional issues about learning data like the class-imbalanced data (Kovashka et al., 2016), label noised data (Natarajan et al., 2013) or out-of-distributed data (Hendrycks & Gimpel, 2017) need further investigated in federated adversarial training. As for the inference threaten, one practical scenario is that different client may face different kinds of adversarial attacks (Carlini & Wagner, 2017; Xiao et al., 2018), whether federated adversarial training can help for gaining various type of robustness on different clients is still under explored.

# B   PROOF

## B.1   PROOF OF EQ. (1) AND THEOREM 4.1

We proof the Eq. (1) and Theorem 4.1 in this section.

Recall the $\alpha$-slack mechanism for the inner-maximization objective decomposition with $K$ independent populations as follows,

$$
\begin{aligned}
\mathcal{L}_{AT} &= \frac{1}{N} \sum_{n=1}^{N} \max_{\tilde{x}_n \in \mathcal{B}_\epsilon[x_n]} \ell(f(\tilde{x}_n), y_n) = \sum_{k=1}^{K} \frac{N_k}{N} \underbrace{\left( \frac{1}{N_k} \sum_{n=1}^{N_k} \max_{\tilde{x}_n \in \mathcal{B}_\epsilon[x_n]} \ell(f(\tilde{x}_n^k), y_n^k) \right)}_{\mathcal{L}_k} \\
&\geq (1+\alpha) \sum_{k=1}^{\widehat{K}} \frac{N_{\phi(k)}}{N} \mathcal{L}_{\phi(k)} + (1-\alpha) \sum_{k=\widehat{K}+1}^{K} \frac{N_{\phi(k)}}{N} \mathcal{L}_{\phi(k)} \quad \text{s.t. } \alpha \in [0, 1), \ \widehat{K} \leq \frac{K}{2} \\
&\doteq \mathcal{L}^\alpha(\widehat{K}),
\end{aligned}
\tag{5}
$$

where $\phi(\cdot)$ is a function which maps the index to the original population group sorted by $\{\frac{N_k}{N}\mathcal{L}_k\}$ in an ascending order. Here $\phi(\cdot)$ is to bind the terms before and after the sort and does not affect the normal gradient back-propagation, since each gradient path of samples is traceable..

*proof of Eq. (1).* The deduction of the inequality in Eq. (5) can be formulated in the following. Given $\alpha \in [0,1)$ and $\widehat{K} \leq \frac{K}{2}$ with the population sorted by $\{\frac{N_k}{N}\mathcal{L}_k\}$ in an ascending order, we have $\sum_{k=1}^{\widehat{K}} \frac{N_{\phi(k)}}{N}\mathcal{L}_{\phi(k)} \leq \sum_{k=\widehat{K}+1}^{K} \frac{N_{\phi(k)}}{N}\mathcal{L}_{\phi(k)}$. Then, we have the following relationship by subtraction,

$$
\begin{aligned}
&\sum_{k=1}^{\widehat{K}} \frac{N_{\phi(k)}}{N} \mathcal{L}_{\phi(k)} + \sum_{k=\widehat{K}+1}^{K} \frac{N_{\phi(k)}}{N} \mathcal{L}_{\phi(k)} - (1+\alpha) \sum_{k=1}^{\widehat{K}} \frac{N_{\phi(k)}}{N} \mathcal{L}_{\phi(k)} - (1-\alpha) \sum_{k=\widehat{K}+1}^{K} \frac{N_{\phi(k)}}{N} \mathcal{L}_{\phi(k)} \\
&= \alpha \cdot \left( \sum_{k=\widehat{K}+1}^{K} \frac{N_{\phi(k)}}{N} \mathcal{L}_{\phi(k)} - \sum_{k=1}^{\widehat{K}} \frac{N_{\phi(k)}}{N} \mathcal{L}_{\phi(k)} \right) \geq 0.
\end{aligned}
\tag{6}
$$

$\square$

*proof of Theorem 4.1.* It can be naturally proved by Eq. (6). If $\alpha_1 > \alpha_2$, then we have,

$$
\begin{aligned}
\mathcal{L}^{\alpha_1}(\widehat{K}) - \mathcal{L}^{\alpha_2}(\widehat{K}) &= (1+\alpha_1) \sum_{k=1}^{\widehat{K}} \frac{N_{\phi(k)}}{N} \mathcal{L}_{\phi(k)} + (1-\alpha_1) \sum_{k=\widehat{K}+1}^{K} \frac{N_{\phi(k)}}{N} \mathcal{L}_{\phi(k)} \\
&\quad - (1+\alpha_2) \sum_{k=1}^{\widehat{K}} \frac{N_{\phi(k)}}{N} \mathcal{L}_{\phi(k)} - (1-\alpha_2) \sum_{k=\widehat{K}+1}^{K} \frac{N_{\phi(k)}}{N} \mathcal{L}_{\phi(k)} \\
&= (\alpha_1 - \alpha_2) \left( \sum_{k=1}^{\widehat{K}} \frac{N_{\phi(k)}}{N} \mathcal{L}_{\phi(k)} - \sum_{k=\widehat{K}+1}^{K} \frac{N_{\phi(k)}}{N} \mathcal{L}_{\phi(k)} \right) \leq 0
\end{aligned}
\tag{7}
$$

Similarly, we can prove $\mathcal{L}^\alpha(\widehat{K}_1) < \mathcal{L}^\alpha(\widehat{K}_2)$ if $\widehat{K}_1 > \widehat{K}_2$. $\square$

## B.2 PROOF OF THEOREM 4.2

Based on the convergence of Adversarial Training (Sinha et al., 2018), we proof Theorem 4.2 in this section. Following (Sinha et al., 2018), we firstly make some required Assumptions B.1 and B.2 and provide the corresponding Lemma B.3 that defines the Lipschitzan condition in Theorem 4.2.

**Assumption B.1.** The function $c : \mathcal{X} \times \mathcal{X} \to \mathbb{R}_+$ is continuous. For each $x_0 \in \mathcal{X}$, $c(\cdot, x_0)$ is $l$-strongly convex with respect to the norm $||\cdot||$.

**Assumption B.2.** Let $||\cdot||_*$ is the dual norm to $||\cdot||$, the loss $\ell : \Theta \times \mathcal{X} \to \mathbb{R}$ satisfies the Lipschitzian smoothness conditions

$$
||\nabla_\theta \ell(\theta; x) - \nabla_\theta \ell(\theta'; x)||_* \leq L_{\theta\theta} ||\theta - \theta'||, \quad ||\nabla_x \ell(\theta; x) - \nabla_x \ell(\theta; x')||_* \leq L_{xx} ||x - x'||,
$$

$$
||\nabla_\theta \ell(\theta; x) - \nabla_\theta \ell(\theta; x')||_* \leq L_{\theta x} ||x - x'||, \quad ||\nabla_x \ell(\theta; x) - \nabla_x \ell(\theta'; x)||_* \leq L_{x\theta} ||\theta - \theta'||.
$$

**Lemma B.3.** *Let $f : \Theta \times \mathcal{X} \to \mathbb{R}$ be differentiable and $\lambda$-strongly concave in $x$ with respect to the norm $||\cdot||$, and define $\bar{f}(\theta) = \sup_{x \in \mathcal{X}} f(\theta, x)$. Let $g_\theta(\theta, x) = \nabla_\theta f(\theta, x)$ and $g_x(\theta, x) = \nabla_x f(\theta, x)$ and assume $g_\theta$ and $g_x$ satisfy Assumption B.2 with $\ell(\theta, x)$ replaced with $f(\theta, x)$. Then $\bar{f}$ is differentiable, and letting $x^*(\theta) = \arg \max_{x \in \mathcal{X}} f(\theta, x)$, we have $\nabla \bar{f}(\theta) = g_\theta(\theta, x^*(\theta))$. Moreover,*

$$||x^*(\theta_1) - x^*(\theta_2)|| \leq \tfrac{L_{x\theta}}{\lambda}||\theta_1 - \theta_2|| \text{ and } ||\nabla \bar{f}(\theta) - \nabla \bar{f}(\theta')||_* \leq (L_{\theta\theta} + \tfrac{L_{\theta x}L_{x\theta}}{\lambda}||\theta - \theta'||).$$

*proof of Theorem 4.2.* Let $c : \mathcal{X} \times \mathcal{X} \to \mathbb{R}_+ \cup \{\infty\}$, where $c(x, x_0)$ is the "cost" for an adversary to perturb $x_0$ to $x$. Let $f(\theta, x; x_0) = \ell(\theta; x) - \gamma c(x, x_0)$, noting that the gradient steps is preformed as $g^t = \nabla_\theta f(\theta^t, \hat{x}; x^t)$, where $\hat{x}$ is an approximate maximizer of $f(\theta, x; x^t)$ in $x$, and $\theta^{t+1} = \theta^t - \mu_t g^t$. We assume $\mu_t \leq \frac{1}{L_\psi}$ in the rest of the proof, which is satisfied for the constant step size $\mu = \sqrt{\frac{\Delta_{\mathcal{L}}}{L_\psi T \delta^2}}$ and $T \geq \frac{L_\psi \Delta_{\mathcal{L}}}{\delta^2}$. By a Taylor expansion using the $L_\psi$-smoothness of the objective $\mathcal{L}_k$ for $k$-th client, we have

$$\mathcal{L}_k\big|_{\theta^{t+1}} \leq \mathcal{L}_k\big|_{\theta^t} + \left\langle \nabla \mathcal{L}_k\big|_{\theta^t}, \theta^{t+1} - \theta^t \right\rangle + \frac{L_\psi}{2}||\theta^{t+1} - \theta^t||_2^2 \tag{8}$$

$$= \mathcal{L}_k\big|_{\theta^t} - \mu_t ||\nabla \mathcal{L}_k\big|_{\theta^t}||_2^2 + \frac{L_\psi \mu^2}{2}||g^t||_2^2 + \mu_t \left\langle \nabla \mathcal{L}_k\big|_{\theta^t}, \nabla \mathcal{L}_k\big|_{\theta^t} - g^t \right\rangle$$

$$= \mathcal{L}_k\big|_{\theta^t} - \mu_t \left(1 - \frac{1}{2}L_\psi \mu^2\right)||\nabla \mathcal{L}_k\big|_{\theta^t}||_2^2$$

$$+ \mu_t(1 - L_\psi \mu)\left\langle \nabla \mathcal{L}_k\big|_{\theta^t}, \nabla \mathcal{L}_k\big|_{\theta^t} - g^t \right\rangle + \frac{L_\psi \mu^2}{2}||g^t - \nabla \mathcal{L}_k\big|_{\theta^t}||_2^2$$

Consider the function $\phi_\gamma(\theta; x_0) = \sup_{x \in \mathbb{Z}} f(\theta, x; x_0)$, we define the potentially biased errors $\zeta^t = g^t - \nabla_\theta \phi_\gamma(\theta^t; x^t)$. Then we have the following relationship,

$$\mathcal{L}_k\big|_{\theta^{t+1}} \leq \mathcal{L}_k\big|_{\theta^t} - \mu_t \left(1 - \frac{1}{2}L_\psi \mu^2\right)||\nabla \mathcal{L}_k\big|_{\theta^t}||_2^2 \tag{9}$$

$$+ \mu_t(1 - L_\psi \mu)\left\langle \nabla \mathcal{L}_k\big|_{\theta^t}, \nabla \mathcal{L}_k\big|_{\theta^t} - \nabla_\theta \phi_\gamma(\theta^t; x^t) \right\rangle$$

$$- \mu_t(1 - L_\psi \mu_t)\left\langle \nabla \mathcal{L}_k\big|_{\theta^t}, \zeta^t \right\rangle + \frac{L_\psi \mu^2}{2}||\nabla_\theta \phi_\gamma(\theta^t; x^t) + \zeta^2 - \nabla \mathcal{L}_k\big|_{\theta^t}||_2^2$$

$$= \mathcal{L}_k\big|_{\theta^t} - \mu_t \left(1 - \frac{1}{2}L_\psi \mu^2\right)||\nabla \mathcal{L}_k\big|_{\theta^t}||_2^2$$

$$+ \mu_t(1 - L_\psi \mu)\left\langle \nabla \mathcal{L}_k\big|_{\theta^t}, \nabla \mathcal{L}_k\big|_{\theta^t} - \nabla_\theta \phi_\gamma(\theta^t; x^t) \right\rangle$$

$$- \mu_t(1 - L_\psi \mu_t)\left\langle \nabla \mathcal{L}_k\big|_{\theta^t}, \zeta^t \right\rangle$$

$$+ \frac{L_\psi \mu_t^2}{2}\left(||\zeta^t||_2^2 + ||\nabla_\theta \phi_\gamma(\theta^t; x^t) - \nabla \mathcal{L}_k\big|_{\theta^t}||_2^2 + 2\left\langle \nabla_\theta \phi_\gamma(\theta^t; x^t) - \nabla \mathcal{L}_k\big|_{\theta^t}, \zeta^t \right\rangle\right).$$

Since $\pm\langle a, b \rangle \leq \frac{1}{2}\left(||a||_2^2 + ||b||_2^2\right)$, we have

$$\mathcal{L}_k\big|_{\theta^{t+1}} \leq \mathcal{L}_k\big|_{\theta^t} - \frac{\mu_t}{2}||\nabla \mathcal{L}_k\big|_{\theta^t}||_2^2 + \mu_t((1 - L_\psi \alpha))\left\langle \nabla \mathcal{L}_k\big|_{\theta^t}, \nabla \mathcal{L}_k\big|_{\theta^t} - \nabla_\theta \phi_\gamma(\theta^t; x^t) \right\rangle \tag{10}$$

$$+ \frac{\mu_t((1 + L_\psi \mu))}{2}||\zeta||_2^2 + L_\psi \mu_t^2||\nabla_\theta \phi_\gamma(\theta^t; x^t) - \nabla \mathcal{L}_k\big|_{\theta^t}||_2^2.$$

Then, letting $x_*^t = \arg \max_x f(\theta^t, x; x^t)$, the error $\zeta^t$ satisfies,

$$||\zeta||_2^2 = ||\nabla_\theta \phi_\gamma(\theta^t; x^t) - \nabla f(\theta, \hat{x}^t; x^t)||_2^2 = ||\nabla_\theta \ell(\theta, x_*^t) - \nabla_\theta \ell(\theta, \hat{x}^t)||_2^2 \tag{11}$$

$$\leq L_{\theta_x}||\hat{x}^t - x_*^t||_2^2 \leq \frac{2L_{\theta_x}^2}{\lambda}\epsilon, \tag{12}$$

where the final inequality utilize the $\lambda = \gamma - L_{xx}$ strong-concavity of $x \mapsto f(\theta, x; x_0)$. For convenience, let $\hat{\epsilon} = \frac{2L_{\theta_x}^2}{\gamma - L_{xx}}\epsilon$. Taking conditional expectations in Eq. (10) and using $\mathbb{E}\left[\nabla_\theta \phi_\gamma(\theta^t; x^t)|\theta^t\right] =$

$\nabla \mathcal{L}_k \big|_{\theta^t}$, we have,

$$\mathbb{E}\left[\mathcal{L}_k\big|_{\theta^{t+1}} - \mathcal{L}_k\big|_{\theta^t}\big|\theta^t\right] \leq -\frac{\mu_t}{2}||\nabla\mathcal{L}_k\big|_{\theta^t}||_2^2 + \frac{\mu_t((1+L_\psi\mu))}{2}\hat{\epsilon} + L_\psi\mu_t^2||\nabla_\theta\phi_\gamma(\theta^t;x^t) - \nabla\mathcal{L}_k\big|_{\theta^t}||_2^2 \tag{13}$$

$$\leq -\frac{\mu_t}{2}||\nabla\mathcal{L}_k\big|_{\theta^t}||_2^2 + \mu_t\hat{\epsilon} + L_\psi\mu_t^2||\nabla_\theta\phi_\gamma(\theta^t;x^t) - \nabla\mathcal{L}_k\big|_{\theta^t}||_2^2$$

Since $\mu_t \leq \frac{1}{L_\psi}$, taking a fixed step size $\mu$, we have,

$$\mathbb{E}\left[||\nabla\mathcal{L}_k\big|_{\theta^t}||_2^2\right] - 2\hat{\epsilon} \leq \frac{2}{\mu}\mathbb{E}\left[\mathcal{L}_k\big|_{\theta^t} - \mathcal{L}_k\big|_{\theta^{t+1}}\right] + 2L_\psi\mu\delta^2 \tag{14}$$

Because $\mathbb{E}\left[||\nabla_\theta\phi_\gamma(\theta;Z) - \nabla\mathcal{L}_k\big|_\theta||_2^2\right] \leq \delta^2$, summing over $t$, we have,

$$\frac{1}{T}\sum_{t=1}^{T}\mathbb{E}\left[||\nabla\mathcal{L}_k\big|_{\theta^t}||_2^2\right] - 2\hat{\epsilon} \leq \frac{2}{\mu T}(\mathcal{L}_k\big|_{\theta^0} - \mathbb{E}[\mathcal{L}_k\big|_{\theta^T}]) + 2L_\psi\mu\delta^2$$

$$\leq \frac{2\Delta}{\mu T} + 2L_\psi\mu\delta^2 \tag{15}$$

Since $\mu = \sqrt{\frac{Delta}{L_\psi T\delta^2}}$, and $\lambda = \gamma - L_{xx}$, we can get the following result,

$$\frac{1}{T}\sum_{t=1}^{T}\mathbb{E}\left[||\nabla\mathcal{L}_k\big|_{\theta^t}||_2^2\right] \leq \frac{4L_{\theta x}^2\epsilon}{\lambda} + 4\delta\sqrt{\frac{L\Delta}{T}} \tag{16}$$

Adopting our $\alpha$-slack mechanism, we have,

$$\frac{1}{T}\sum_{t=1}^{T}\mathbb{E}\left[\left|\left|\nabla\mathcal{L}^\alpha(\widehat{K})\big|_{\theta^t}\right|\right|_2^2\right]$$

$$= \frac{1}{T}\sum_{t=1}^{T}\mathbb{E}\left[\left|\left|(1+\alpha)\sum_{k=1}^{\widehat{K}}\frac{N_{\phi(k)}^{(t)}}{N}\nabla\mathcal{L}_{\phi(k)}\big|_{\theta^t} + (1-\alpha)\sum_{k=\widehat{K}+1}^{K}\frac{N_{\phi(k)}^{(t)}}{N}\nabla\mathcal{L}_{\phi(k)}\big|_{\theta^t}\right|\right|_2^2\right]$$

$$\leq \frac{1}{T}\sum_{t=1}^{T}\left((1+\alpha)\sum_{k=1}^{\widehat{K}}\frac{N_{\phi(k)}^{(t)}}{N}\mathbb{E}\left[\left|\left|\nabla\mathcal{L}_{\phi(k)}\big|_{\theta^t}\right|\right|_2^2\right] + (1-\alpha)\sum_{k=\widehat{K}+1}^{K}\frac{N_{\phi(k)}^{(t)}}{N}\mathbb{E}\left[\left|\left|\nabla\mathcal{L}_{\phi(k)}\big|_{\theta^t}\right|\right|_2^2\right]\right)$$

$$\leq \frac{1}{T}\sum_{t=1}^{T}\left((1+\alpha)\sum_{k=1}^{\widehat{K}}\frac{N_{\phi(k)}^{(t)}}{N} + (1-\alpha)\sum_{k=\widehat{K}+1}^{K}\frac{N_{\phi(k)}^{(t)}}{N}\right)\left(\frac{4L_{\theta x}^2\epsilon}{\lambda} + 4\delta\sqrt{\frac{L\Delta}{T}}\right)$$

$$= \frac{1}{T}\sum_{t=1}^{T}\left(1 + \alpha\frac{\sum_{k=1}^{\widehat{K}}N_{\phi(k)}^{(t)} - \sum_{k=\widehat{K}+1}^{K}N_{\phi(k)}^{(t)}}{N}\right)\left(\frac{4L_{\theta x}^2\epsilon}{\lambda} + 4\delta\sqrt{\frac{L\Delta}{T}}\right)$$

$$= \left(1 + \alpha\frac{\frac{1}{T}\sum_{t=1}^{T}\xi^{(t)}}{N}\right)\left(\frac{4L_{\theta x}^2\epsilon}{\lambda} + 4\delta\sqrt{\frac{L\Delta}{T}}\right), \tag{17}$$

where $\xi^{(t)} = \sum_{k=1}^{\widehat{K}}N_{\phi(k)}^{(t)} - \sum_{k=\widehat{K}+1}^{K}N_{\phi(k)}^{(t)}$ to simplify the notations. $\qquad\square$

## B.3 Proof of Theorem 4.3

Based on the convergence of FedAvg (Li et al., 2019), we proof Theorem 4.3 in this section.

First, we make the following assumptions and present some useful lemmas. Specifically, we make the following assumptions. Assumption B.4 and B.5 are standard (typical examples are the $\ell_2$-norm regularized linear regression, logistic regression, or softmax classifier). Assumption B.6 and B.7 have been made by the previous works (Zhang et al., 2013; Li et al., 2019).

**Assumption B.4.** $\mathcal{L}_1, \ldots, \mathcal{L}_K$ are all L-smooth: for all $v$ and $w$, $\mathcal{L}_k(v) \leq \mathcal{L}_k(w) + (v - w)^T \nabla \mathcal{L}_k(w) + \frac{L}{2}||v - w||_2^2$.

**Assumption B.5.** $\mathcal{L}_1, \ldots, \mathcal{L}_K$ are all $\lambda$-strongly convex: for all $v$ and $w$, $\mathcal{L}_k(v) \geq \mathcal{L}_k(w) + (v - w)^T \nabla \mathcal{L}_k(w) + \frac{\lambda}{2}||v - w||_2^2$.

**Assumption B.6.** Let $\xi_t^k$ be sampled from the $k$-th device's local data uniformly at random. The variance of stochastic gradients in each device is bounded: $\mathbb{E}||\nabla \mathcal{L}_k(w_t^k, \xi_t^k) - \nabla \mathcal{L}_k(w_t^k)||^2 \leq \delta_k^2$ for $k = 1, \cdots, K$.

**Assumption B.7.** The expected squared norm of stochastic gradients is uniformly bounded, i.e., $\mathbb{E}||\nabla \mathcal{L}_k(w_t^k, \xi_t^k)||^2 \leq \varsigma^2$ for all $k = 1, \cdots, K$ and $t = 1, \cdots, T - 1$.

We use the following lemmas proved by Li et al. (2019). Let $\theta_t^k$ denotes the model parameter maintained in the $k$-th client at $t$-th step, $\Theta$ represents an immediate result of one step SGD update from $\theta_t^k$. For convenience, we define $\bar{\Theta}_t = \sum_{k=1}^K \frac{N_k}{N} \Theta_t$, $\bar{\theta}_t == \sum_{k=1}^K \frac{N_k}{N} \theta_t$, $\bar{g}_t = \sum_{k=1}^K \frac{N_k}{N} \nabla \mathcal{L}_k(\theta_t^k)$ and $g_t = \sum_{k=1}^K \frac{N_k}{N} \nabla \mathcal{L}_k(\theta_t^k, \xi_t^k)$. Therefore, $\mathbb{E}g_t = \bar{g}_t$.

**Lemma B.8** (Results of one step SGD). *Assume Assumption B.4 and B.5. If $\eta_t \leq \frac{1}{4L}$, we have*

$$\mathbb{E}||\bar{\Theta}_{t+1} - \theta^*||^2 \leq (1 - \eta_t \lambda)\mathbb{E}||\bar{\theta}_t - \theta^*||^2 \tag{18}$$

$$+ \eta_t^2 \mathbb{E}||g_t - \bar{g}_t||^2 + 6L\eta_t^2 \Gamma + 2\mathbb{E} \sum_{k=1}^K \frac{N_k}{N}||\bar{\theta}_t - \theta_t^k||^2,$$

*where $\Gamma = \mathcal{L}^* - \sum_k^K \mathbb{E}_t \frac{N_k}{N} \mathcal{L}_k^* \geq 0$*

**Lemma B.9** (Bounding the variance). *Assume Assumption B.6. It follows that*

$$\mathbb{E}||g_t - \bar{g}_t||^2 \leq \sum_{k=1}^K \left(\frac{N_k}{N}\right)^2 \delta_k^2, \tag{19}$$

**Lemma B.10** (Bounding the divergence of $\theta_t^k$). *Assume Assumption B.7, that $\eta_t$ is non-increasing and $\eta \leq 2\eta_{t+E}$ for all $t > 0$. It follows that*

$$\mathbb{E}\left[\sum_{k=1}^K \frac{N_k}{N}||\bar{\theta}_t - \theta_t^k||^2\right] \leq 4\eta_t^2(E-1)^2 \varsigma^2 \tag{20}$$

*proof of Theorem 4.3.* Let $\Delta_t = \mathbb{E}||\theta_t - \theta^*||^2$. From Lemma B.8, Lemma B.9 and Lemma B.10, it follows that

$$\Delta_{t+1} \leq (1 - \eta_t \lambda)\Delta_t + \eta_t^2 B, \tag{21}$$

where,

$$B = \sum_{k=1}^K \left(\frac{P_k^{(T)}}{1 + \alpha \frac{\xi^{(T)}}{N}}\right)^2 \left(\frac{N_k}{N} \delta_k\right)^2 + 6L\left(\mathcal{L}^* - \sum_{k=1}^K \frac{P_k^{(T)}}{1 + \alpha \frac{\xi^{(T)}}{N}} \frac{N_k}{N} \mathcal{L}_k^*\right) + 8(E-1)^2 \varsigma^2, \tag{22}$$

and $\left(\frac{P_k^{(T)}}{1 + \alpha \frac{\xi^{(T)}}{N}}\right)^2$ and $\frac{P_k^{(T)}}{1 + \alpha \frac{\xi^{(T)}}{N}}$, are acted as the scalar timing by the personalized variance bound $\delta_k^2$ and the local optimum $\mathcal{L}_k^*$ of each client.

For a diminishing stepsize, $\eta_t = \frac{\beta}{\gamma + t}$ for some $\beta > \frac{1}{\lambda}$ and $\gamma > 0$ such that $\eta_1 \leq \min\{\frac{1}{\lambda}, \frac{1}{4L}\} = \frac{1}{4L}$ and $\eta_t \leq 2\eta_{t+E}$. We will prove that $\Delta_t \leq \frac{\nu}{\gamma + t}$, where $\nu = \max\{\frac{\beta^2 B}{\beta\lambda - 1}, (\gamma + 1)\Delta_1\}$. The above can be proved by induction. Firstly, the definition of $\nu$ ensures that it holds for $t = 1$. Assume the conclusion holds for some $t$, it follows that,

$$\Delta_{t+1} \leq (1 - \eta_t \lambda)\Delta_t + \eta_t^2 B$$

$$\leq (1 - \frac{\beta\lambda}{t + \gamma})\frac{\nu}{t + \gamma} + \frac{\beta^2 B}{(t + \gamma)^2}$$

$$= \frac{t + \gamma - 1}{(t + \gamma)^2}\nu + \left[\frac{\beta^2 B}{(t + \gamma)^2} - \frac{\beta\lambda - 1}{(t + \gamma)^2}\nu\right] \tag{23}$$

$$\leq \frac{\nu}{t + \gamma + 1}.$$

Then by the L-smoothness of $\mathcal{L}(\cdot)$,

$$\mathbb{E}[\mathcal{L}_t] - \mathcal{L}^* \le \frac{L}{2}\Delta_t \le \frac{L}{2}\frac{\nu}{\gamma+t} \tag{24}$$

Specifically, if we choose $\beta = \frac{2}{\lambda}, \gamma = \max\{8\frac{L}{\lambda}, E\} - 1$ and denote $\kappa = \frac{L}{\lambda}$, then $\eta_t = \frac{2}{\lambda}\frac{1}{\gamma+t}$. One can verify that the choice of $\eta_t$ satisfies $\eta_t \le 2\eta_{t+E}$ for $t \ge 1$. Then we have

$$\nu = \max\left\{\frac{\beta^2 B}{\beta\lambda-1}, (\gamma+1)\Delta_1\right\} \le \frac{\beta^2 B}{\beta\lambda-1} + (\gamma+1)\Delta_1 \le \frac{4B}{\lambda^2} + (\gamma+1)\Delta_1, \tag{25}$$

and

$$\mathbb{E}[\mathcal{L}_t] - \mathcal{L}^* \le \frac{L}{2}\frac{\nu}{\gamma+t} \le \frac{\kappa}{\gamma+t}\left(\frac{2B}{\lambda} + \frac{\lambda(\gamma+1)}{2}\Delta_1\right) \tag{26}$$

$\square$

## C FURTHER DISCUSSION ABOUT FIGURE 2

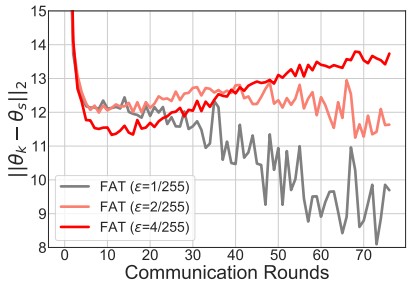
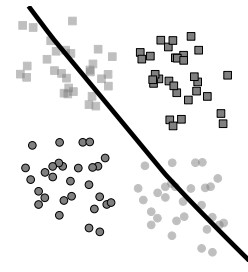
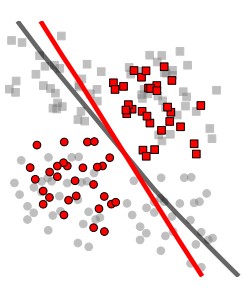

(a) Client drift with different adv. generation      (b) w/o adv. generation      (c) w/ adv. generation

Figure 5: (a): the client drift (Li et al., 2018) with different adv. generation varied from weak (i.e., $\epsilon = 1/255$) to strong (i.e., $\epsilon = 4/255$) on SVHN. $\|\theta_k - \theta_s\|_2$ indicates the parameter difference between local model and averaged global model. It verified that stronger adv. generation leads to intensified heterogeneity. (b)-(c): The decision boundary of a binary classification task on one local client. The black squares and circles indicate samples of two classes in the client, the light gray ones indicate that of the other client. The red squares and circles indicate the generated adversarial data. It serves as a potential explanation for the bias exacerbation caused by local adversarial generation. Note that, (b)-(c) are obtained by training the neural networks.

In this section, we provide a verification about intensified heterogeneity using the real dataset, i.e., SVHN (Netzer et al., 2011) (in Figure 5(a)), and provide a two-dimensional result (in Figures 5(b) and 5(c)) to explain the exacerbated optimization bias induced by the local adversarial generation.

First, we conduct the experiments under Non-IID setting to verify the relationship of intensified heterogeneity with adversarial generation in the SVHN dataset, which is for the multi-classification task. To be specific, we compare the FAT on 5 clients using different adversarial strength to show the effect of adversarial generation on local clients. Other setups keep the same with that of Table 3.

In Figure 5(a), we conduct FAT using the adversarial generation of different $\epsilon$-ball (from weak $1/255$ to strong $4/255$) and check the client drift, which has been formally defined in Karimireddy et al. (2020) to show the optimization bias in federated learning. The client drift is widely adopted in federated learning literature (Smith et al., 2017; Li et al., 2018; 2021b; Karimireddy et al., 2020)] to reflect the degree of heterogeneity (Zhao et al., 2018; Li et al., 2019). The statics of client drift with different adversarial generation can indicate the relationship between inner-maximization with the intensified heterogeneity. Through the results, we can find that as adversarial strength increases, the heterogeneity is also exacerbated along with training. The similar trend can be also found in the experiments with other dataset in Figure 10 of Appendix E.4, where we also present the corresponding robust accuracy. This verifies the issue discussed in our motivation part (in Section 4.1) using the real classification task, and it is also consistent with our binary illustration.

Second, we conduct the experiments on a synthetic dataset to present the potential explanation about the heterogeneity exacerbation. To be specific, we conduct basic federated learning and FAT using two clients on the synthetic dataset and check the decision boundary in a two-dimensional plane. It shows how the adversarial generation on local clients can exacerbate the heterogeneity.

As for the experiments on the synthetic dataset, we create its training data randomly using Numpy and train using a neural network with one hidden layer and one Relu activation layer for each clients.

In Figures 5(b) and 5(c), we plot the decision boundary of the training results on a synthetic dataset for FAT using two clients. The Figure 5(b) shows standard training (using the natural data) on the client's local data while the Figure 5(c) shows adversarial training (generating the adversarial data) on the client's local data. Note that, without the training data of other clients, the adversarial generation on the current biased model is highly biased to the local distribution, which even exacerbated the optimization bias and would result in a large client drift or variance, e.g., $\delta^2$ in our theorems. More empirical evidence on CIFAR-10 dataset can also refer to Appendixes E.4 and E.6 which show the statics corresponding the accuracy during the training process.

Table 5: Robust deterioration w.r.t adversarial training strength in FAT.

| Setting | | Performance gap between best and last epoch | | |
|---|---|---|---|---|
| Dataset/Adv. Strength | 2/255 | 4/255 | | 8/255 |
| CIFAR-10 \| FAT | 6.33 | 11.16 | | **13.06** |
| Dataset/Adv. Strength | 1/255 | 2/255 | | 4/255 |
| SVHN \| FAT | 4.65 | 5.63 | | **6.43** |

To better support our conjecture, we compare and robust deterioration in FAT with different adversarial training strength (using the same robust evaluation strength) and summarized the results in Table 5. The results show that more stronger adversarial generation (i.e., the larger inner-maximization) lead to severe robustness deterioration (indicated by the large accuracy gap between the best and last stage). It confirms the rationality of our derived lower bound by $\alpha$-slack mechanism to pursue the adversarial robustness and alleviate the heterogeneity exacerbation. In our Appendix E.4, we also provide more empirical verification that quantitatively measure the intensified heterogeneity, which show that our SFAT indeed alleviate the intensified heterogeneity and achieve the better robust accuracy.

# D   DETAILED ALGORITHM AND LEARNING FRAMEWORK

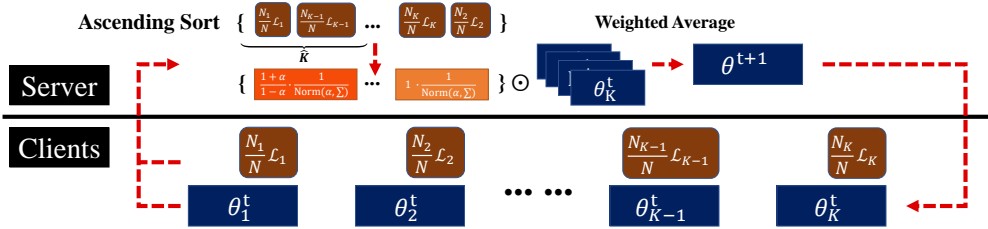

Figure 6: A brief illustration of our Slack Federated Adversarial Training (SFAT) framework. On the client-side, each client will conduct adversarial training on its local data and update the optimized model parameter (*i.e.*, $\theta_k$) with the adversarial training loss (*i.e.*, $\frac{N_k}{N}\mathcal{L}_k$)). On the server-side, after collecting the model parameters and the loss value (information about the training status), the server will conduct an ascending sort and aggregate the global model with a weighted average (denoted by $\odot$) which upweights the top populations of the small-loss client's model parameters with $\alpha$.

In this section, we provide the detailed algorithm (i.e., Algorithm 1) for our Slack Federated Adversarial Training (SFAT) and an intuitive illustration of our proposed SFAT in Figure 6.

---

**Algorithm 1** Slack Federated Adversarial Training

---

**Input:** client number: $K$, communication rounds: $T$, local training epochs per round: $E$, initial server's model parameter: $\theta^0$, hyper-parameter for aggregation: $\alpha$, number of enhanced clients: $\widehat{K}$;
**Output:** globally robust model $\theta^T$;

1: **for** t $= 1, \ldots, T$ **do**
2:     **Clients:** [ perform adversarial training]
3:     **for** client $k = 1, \ldots, K$ **do**
4:        $\theta_k^t, \mathcal{L}_k = \text{AT}(\theta_k^t, E)$ (Madry et al., 2018)
5:     **end for**
6:     **Server:** [ performs slacked aggregation]
7:     $\mathcal{L}_{all} \leftarrow [\frac{N_1}{N}\mathcal{L}_1, \frac{N_2}{N}\mathcal{L}_2, \ldots, \frac{N_K}{N}\mathcal{L}_K], \quad \mathcal{L}_{\text{sorted}} \leftarrow \text{Ascending\_Sort}(\mathcal{L}_{all})$;
8:     $\forall k, \ P_k = (\frac{1+\alpha}{1-\alpha} \cdot \mathbb{1}(\frac{N_k}{N}\mathcal{L}_k \leq \mathcal{L}_{\text{sorted}}[\widehat{K}]) + 1 \cdot \mathbb{1}(\frac{N_k}{N}\mathcal{L}_k > \mathcal{L}_{\text{sorted}}[\widehat{K}]))/((\sum_{k=1}^{K} P_k) + \frac{2\alpha}{1-\alpha})$;
9:     $\theta^{t+1} = \frac{1}{\sum_{k=1}^{K} N_k} \sum_{k=1}^{K} P_k N_k \theta_k^t$;
10: **end for**

---

For simplifying the practical use and adaptation, we can mainly assign higher weights for the client having smaller adversarial training losses to realize the relative weighting illustrated in Figure 6. To be specific, we can set the normalized $P_k = ((1+\alpha)/(1-\alpha) \cdot \mathbb{1}(\frac{N_k}{N}\mathcal{L}_k \leq \mathcal{L}_{\text{sorted}}[\widehat{K}]) + 1 \cdot \mathbb{1}(\frac{N_k}{N}\mathcal{L}_k > \mathcal{L}_{\text{sorted}}[\widehat{K}]))/((\sum_{k=1}^{K} P_k) + 2\alpha/(1-\alpha))$ in the aggregation to ensure the expected lower bound.

Based on the $\alpha$-slack mechanism, we provide a new framework for the combination of adversarial training with federated learning. It is orthogonal to a variety of different adversarial training (Zhang et al., 2019; Alayrac et al., 2019; Wang et al., 2020a; Jiang et al., 2020; Chen et al., 2021; Carmon et al., 2019; Madry et al., 2018; Chen et al., 2020; Ding et al., 2020; Li et al., 2021c; Chen et al., 2021; 2022) methods and federated optimization algorithms (McMahan et al., 2017; Li et al., 2018; 2021b; Kairouz et al., 2019) which pursue the adversarial robustness or alleviate the data heterogeneity as well as other specific issues on the client side, and for the specific practical challenge for federated settings (Shah et al., 2021; Hong et al., 2021). Those all can be flexibly adopted into our framework or extend to fit other training constraints (Kairouz et al., 2019) of federated learning.

## E EXPERIMENTAL DETAILS AND MORE COMPREHENSIVE RESULTS

In this section, we first provide the details about our experimental setups for dataset, training and evaluation. Then we provide more comprehensive results for better understanding the characteristics of our SFAT and the performance verification in different settings. In Appendix E.1, we tracing and discuss the dynamics of our SFAT on choosing the upweighted clients. In Appendix E.2, we present the results of training with different local epochs. In Appendix E.3, we present and discuss the comparison of Re-SFAT v.s. SFAT. In Appendixes E.4 and E.6, we report the client drift and variance with the corresponding robust accuracy during training. In Appendix E.8, we discuss the orthogonal effects of SFAT on intensified heterogeneity. In Appendix E.9, we verify SFAT using more clients. In Appendix E.10, we verify SFAT using unequal data splits under Non-IID setting. In Appendix E.11, we verify SFAT in a more practical real-world situation. In Appendix E.12, we report the performance results on both Non-IID and IID setting across the three benchmarked datasets.

**Dataset.** We conduct the experiments on three benchmark datasets, *i.e., SVHN* (Netzer et al., 2011), *CIFAR-10* and *CIFAR-100* (Krizhevsky, 2009) as well as a real-world dataset *CelebA* (Caldas et al., 2018) for federated adversarial training. For the IID scenario, we randomly distribute these datasets to each client. For simulating the Non-IID scenario, we follow McMahan et al. (2017); Shah et al. (2021) to distribute the training data based on their labels. To be specific, a skew parameter $s$ is utilized in the data partition introduced by Shah et al. (2021), which enables $K$ clients to get a majority of the data samples from a subset of classes. We denote the set of all classes in a dataset as $\mathcal{Y}$ and create $\mathcal{Y}_k$ by dividing all the class labels equally among $K$ clients. Accordingly, we split the data across $K$ clients that each client has $(100 - (K-1) \times s)\%$ of data for the class in $\mathcal{Y}_k$ and $s\%$ of data in other split sets. In most experiments, we set $s = 2$ for simulating the Non-IID partition with 5 clients as Shah et al. (2021) recommended.

Table 6: Brief summary of the basic experimental details about SFAT

| Dataset | Network | Local epochs | $K$ | $\widehat{K}$ |
|---|---|---|---|---|
| *CIFAR-10* | NIN (Shah et al., 2021) | 10 | 5 | 1 |
| *CIFAR-100* | ResNet-18 (Chen et al., 2021) | 3 | 20 | 4 |
| *SVHN* | SmallCNN (Zhang et al., 2019) | 2 | 5 | 1 |

**Training and evaluation.**   In the experiments, we follow the previous works (Zhang et al., 2019; Shah et al., 2021) to leverage the same architectures, *i.e., NIN* (Lin et al., 2014) for *CIFAR-10*, *ResNet-18* (He et al., 2016) for *CIFAR-100* and *Small CNN* (Zhang et al., 2019) for *SVHN*.

For the local training batch size, we set 32 for *CIFAR-10*, 128 for *CIFAR-100* and *SVHN*. For the training schedule, SGD is adopted with 0.9 momentum for 100 communication rounds under 5 clients as in (Hong et al., 2021; Shah et al., 2021), and the weight decay = 0.0001. For adversarial training, we set the configurations of PGD respectively (Madry et al., 2018) for different datasets. On *CIFAR-10/CIFAR-100*, we set the perturbation bound $\epsilon = 8/255$, the PGD step size $2/255$ and set the PGD step number 10. On *SVHN*, we set the perturbation bound $\epsilon = 4/255$, the PGD step size $1/255$. The PGD generation for all the datasets keep the same step number 10.

Regarding the evaluation, the accuracy for the natural test data and that for the adversarial test data are computed following Madry et al. (2018); Zhang et al. (2019). Note that, the adversarial test data are generated by FGSM, PGD-20, C&W  (Carlini & Wagner, 2017) attack with the same perturbation bound and step size as the training. All the adversarial generations have a random start, i.e, the uniformly random perturbation of $[-\epsilon, \epsilon]$ added to the natural data before attacking iterations. Besides, we also report the robustness under a stronger AutoAttack, termed as AA for simplicity. All the experiments are conducted for multiple times using NVIDIA Tesla V100-SXM2.

As for our SFAT, different training tasks adopt different $\alpha$-slack mechanism considering different characteristic of local training data, specifically, we set $\alpha = 1/6$ (*i.e.,* $\frac{1+\alpha}{1-\alpha} = 1.4$) for the experiments on *CIFAR-10*, and $\alpha = 1/11$ (*i.e.,* $\frac{1+\alpha}{1-\alpha} = 1.2$) for the experiments on *CIFAR-100* and $\alpha = 1/11$ (*i.e.,* $\frac{1+\alpha}{1-\alpha} = 1.2$) for the experiments on *SVHN*. As for FedProx, we set its original hyper-parameter $\mu = 0.01$ for each dataset and the $\alpha$ for our $\alpha$-slack mechanism are $1/11$, $1/11$, and $1/11$. As for Scaffold, the $\alpha$ adopted for previous datasets are $1/11$, $3/23$ and $1/11$ correspondingly.

As for the choice of the hyper-parameter $\alpha$, one useful way to set it might be progressively probing its effect in a value-growth manner. When the $\alpha$ is very small, the objective will approximately degenerate the original objective of FAT, so does the performance with no harm. Slightly enlarging $\alpha$ can improve the performance due to the benefit on alleviating the intensified heterogeneity during aggregation, and then make a stop in one point where the performance becomes drop.

### E.1   THE DYNAMIC OF SFAT

In this part, we present the dynamics of our SFAT about the critical $\alpha$-slack mechanism. To be specific, we visualize the selected clients in each communication rounds for our slack aggregation.

In one experiment on *CIFAR-10* (Non-IID), we trace the index of top client (i.e., the selected client that is upweighted in our mechanism) and find the top weight dynamically routes among different clients instead of the single or a subset of 5 clients. The empirical results confirmed there is no dominant client exists during the training. To further check whether our SFAT will result in unfair attention to different clients, we investigate the difference of client's training accuracy, the accuracy gap (32.40%) of the best client with the worst client is comparable with that (32.23%) in SFAT. Intuitively, since the large adversarial training loss can automatically balance the client-wise aggregation weight, there is no unfair attention to exacerbate the performance difference among clients in our experiments, which are verified by the above gap and the similar assignments for the weighting index.

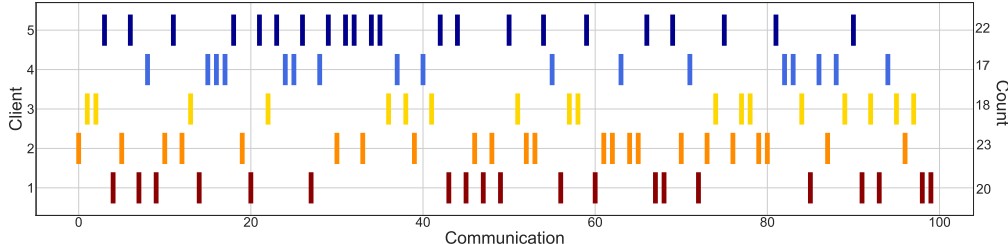

Figure 7: The index of the top-$\widehat{K}$ clients with the small losses in SFAT ($\alpha = 1/6$, $\widehat{K} = 1$) in each communication round on *CIFAR-10* (Non-IID) as well as the total account. We can see that it is dynamically routing among all clients instead of fixed, and each client has similar assignments.

**Robust performance on each client.** In Figure 7, we can find there is no dominant client exists, which means that all clients are ever chose to be up-weighted/down-weighted during the training. Thus, our SFAT explores to help all clients to be better. Besides, the global model is redistributed to each client in each communication round of federated learning, which can also avoid the bias accumulation for each single client. To verify each client performance in our experiments, we report the robust training accuracy of each client in our experiments and summarize in Table 7. The results show the training performance of each client is not worse than the original FAT.

Table 7: Robust training accuracy on each client w.r.t. different methods.

| Setting | | Client | | | | |
|---|---|---|---|---|---|---|
| Method/Client | | 1 | 2 | 3 | 4 | 5 |
| CIFAR-10 | FAT | 59.35 | 65.97 | 75.96 | 77.12 | 80.37 |
| | **SFAT** | 60.05 | 66.57 | 77.17 | 78.79 | 83.23 |
| SVHN | FAT | 89.50 | 82.12 | 89.60 | 81.96 | 87.88 |
| | **SFAT** | 91.69 | 85.58 | 91.93 | 85.31 | 90.35 |

To further check the generalization performance on each client, we also conduct an extra experiment where we split a small part of local data in each client to serve as the local test data and evaluate their training and test performance using FAT and the proposed SFAT on SVHN dataset. According to the summarized results in Table 8, the generalization performance (indicated by the gap of robust training and test accuracy) of our SFAT are also generally better than FAT in SVHN dataset.

Table 8: Robust performance gap on each client w.r.t. different methods.

| Setting | | | Client | | | | |
|---|---|---|---|---|---|---|---|
| Method/Client | | | 1 | 2 | 3 | 4 | 5 |
| SVHN | FAT | Train | 87.51 | 88.19 | 85.91 | 78.99 | 79.17 |
| | | Test | 72.96 | 70.64 | 64.30 | 61.97 | 66.57 |
| | | Gap | 14.55 | 17.55 | 21.61 | 17.02 | 12.65 |
| | SFAT | Train | 87.59 | 88.41 | 85.97 | 79.27 | 79.04 |
| | | Test | 73.65 | 71.46 | 65.80 | 63.48 | 66.62 |
| | | Gap | 13.94 | 16.95 | 20.17 | 15.79 | 12.42 |

## E.2 EXPERIMENTS WITH DIFFERENT LOCAL TRAINING EPOCHS

In this part, we first report the robust test curve on *CIFAR-10* dataset to compare the FAT and SFAT using different local training epoch in each client. Then we focus on a extreme setup, i.e., 1 local epoch for each client in both *CIFAR-10* and *SVHN* datasets and report the performance comparison, although using 1 local epoch is not practical considering the heavy communication cost it introduced in real-world application (McMahan et al., 2017).

In Figure 8, we conduct the experiments on *CIFAR-10* with different local training epochs. We find FAT exhibits robust deterioration across different settings that impedes further progress towards adversarial robustness of the federated system, and simply changing the local training epoch can not

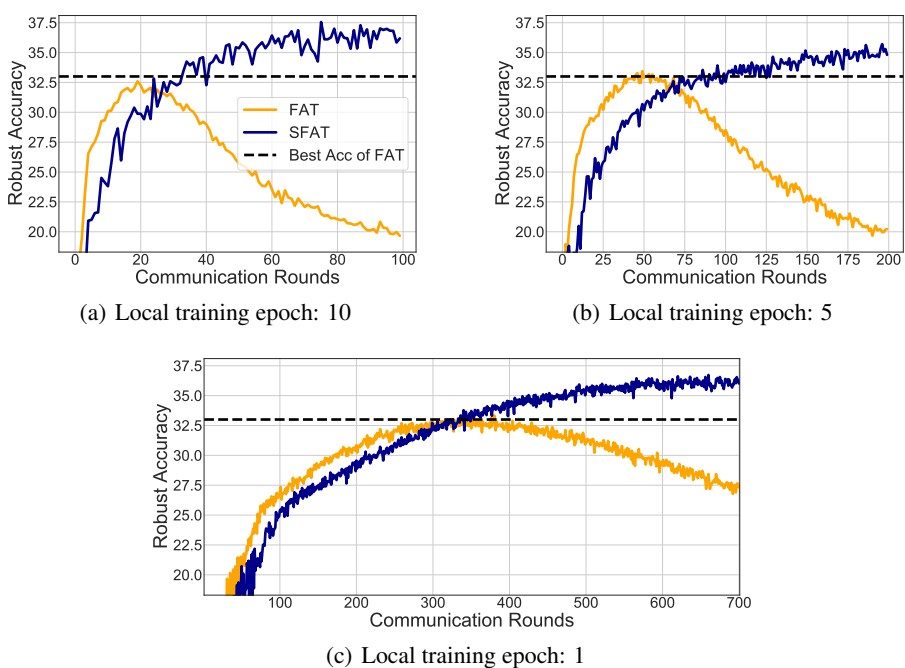

(a) Local training epoch: 10

(b) Local training epoch: 5

(c) Local training epoch: 1

Figure 8: Comparison between FAT and SFAT with different local training epochs. All the experiments are conducted on *CIFAR-10* dataset (Non-IID) with 5 clients, and use PGD-20 (Madry et al., 2018) to evaluate the robust accuracy.

achieve the significantly higher robust accuracy as the dashed black line denoted. In comparison, our SFAT can consistently achieve a higher robust accuracy than FAT by alleviating the deterioration.

Even on the extreme setup, i.e., using 1 local epoch for each client, we can find the intensified heterogeneity also exist since the adversarial generation will be conducted in each optimization step, and can be further mitigate by our SFAT. Reducing the local epoch indeed can alleviate the robustness deterioration since it can reducing the original client drift (McMahan et al., 2017; Li et al., 2018). As the original client drift is alleviated, the adversarial generation will inherit and exacerbate less heterogeneity. However, it does not achieve the nature of this issue and this experimental setup adjustment has similar essence with those federated optimization methods (Li et al., 2018; Karimireddy et al., 2020). The results help us to better understand the effects of the proposed SFAT on combating the intensified heterogeneity via relaxing the inner-maximization to a lower bound.

Table 9: Comparison using FedAvg on CIFAR-10 and SVHN datasets using different local epochs.

| Dataset | Local epochs | Methods | Natural | FGSM | PGD-20 | CW$_\infty$ |
|---------|--------------|---------|---------|------|--------|-------------|
| CIFAR-10 | 10 (in Table 3) | FAT | 57.45% | 39.44% | 32.58% | 30.52% |
| | | **SFAT** | **63.44%** | **45.13%** | **37.17%** | **33.99%** |
| | 1 | FAT | 57.61% | 40.27% | 33.18% | 32.16% |
| | | **SFAT** | **64.85%** | **45.55%** | **37.04%** | **34.84%** |
| SVHN | 2 (in Table 3) | FAT | 91.24% | 87.95% | 68.87% | 67.39% |
| | | **SFAT** | **91.25%** | **88.28%** | **71.72%** | **69.79%** |
| | 1 | FAT | 91.21% | 87.99% | 69.35% | 68.28% |
| | | **SFAT** | **91.98%** | **88.99%** | **72.34%** | **71.05%** |

In Table 9, we report and compare the results under the 1-epoch local training compared with our previous results under multiple epochs used in Table 3. The results verify the consistent effectiveness

of SFAT. Reducing the local epochs to 1 indeed can improve the performance ( 0.5%) of the baseline FAT, but might not be very practical due to the resulting large cost of the communication in federated learning (McMahan et al., 2017). Even on the 1-epoch setting, as shown in Figure 8, the FAT still suffer from the robust deterioration. The results in Table 9 demonstrate that the intensified heterogeneity could be better alleviated by our SFAT. It is similar to our discussion (in Appendix E.8) about the orthogonal effects with federated optimization methods (Li et al., 2018).

### E.3 SFAT VS. RE-SFAT

Table 10: Comparison with emphasize/de-emphasize the client with smallest loss.

| Setting | | | Non-IID | | | |
|---|---|---|---|---|---|---|
| CIFAR-10 | | | Natural | FGSM | PGD-20 | CW$_\infty$ |
| FedAvg | **SFAT:** $\frac{1+\alpha}{1-\alpha} = 1.4$ | emphasize | **63.44%** | **45.13%** | **36.17%** | **33.99%** |
| | **SFAT:** $\frac{1+\alpha}{1-\alpha} = 1.2$ | emphasize | 62.26% | 44.08% | 35.83% | 33.31% |
| | FAT: $\frac{1+\alpha}{1-\alpha} = 1.0$ | original | 57.45% | 39.44% | 32.58% | 30.52% |
| | **Re-SFAT:** $\frac{1+\alpha}{1-\alpha} = 0.8$ | de-emphasize | 50.45% | 34.34% | 27.86% | 26.62% |
| | **Re-SFAT:** $\frac{1+\alpha}{1-\alpha} = 0.6$ | de-emphasize | 40.47% | 28.81% | 24.36% | 23.19% |
| SVHN | | | Natural | FGSM | PGD-20 | CW$_\infty$ |
| FedAvg | **SFAT:** $\frac{1+\alpha}{1-\alpha} = 1.4$ | emphasize | 90.60% | 87.75% | **73.12%** | **70.51%** |
| | **SFAT:** $\frac{1+\alpha}{1-\alpha} = 1.2$ | emphasize | **91.25%** | **88.28%** | 71.72% | 69.79% |
| | FAT: $\frac{1+\alpha}{1-\alpha} = 1.0$ | original | 91.24% | 87.95% | 68.87% | 67.89% |
| | **Re-SFAT:** $\frac{1+\alpha}{1-\alpha} = 0.8$ | de-emphasize | 90.03% | 86.12% | 64.35% | 64.32% |
| | **Re-SFAT:** $\frac{1+\alpha}{1-\alpha} = 0.6$ | de-emphasize | 89.46% | 84.80% | 58.64% | 58.96% |

(a) Robust accuracy      (b) Client drift      (c) Zoom-in Client drift

Figure 9: The robust accuracy w.r.t. the client drift (Li et al., 2018) of Re-SFAT, FAT, and SFAT during training. It shows that Re-SFAT further enhances the intensified heterogeneity as well as the robustness deterioration via emphasizing the client with larger adversarial loss, which is reversed to the operation of our SFAT. This verifies the rationality of SFAT that slack the original objective.

In this part, we start with the experiments about comparing the performance of oppositely using the $\alpha$-slack mechanism, i.e., Re-SFAT, with our original SFAT. Then we discuss its underlying reason as well as the relationship of loss and the intensified heterogeneity.

In Table 10, we conduct an empirical comparison between FAT, SFAT (which *emphasizes* the client model with the smallest adversarial training loss) and Re-SFAT (which is a contrary variant of SFAT that *de-emphasizes* the client model with the smallest adversarial training loss). Here the Re-SFAT share the same spirit with the AFL (Mohri et al., 2019), which seeks to improve the fairness and generalization through a loss-maximization reweighting. The experimental setups keep the same as Table 3 using 5 clients. Both SFAT and Re-SFAT keep the $\widehat{K} = 1$ in all the trails.

Through the results across two benchmarked datasets (i.e., *CIFAR-10* and *SVHN*), We find that de-emphasizing the client with smallest adversarial loss (relatively emphasize those with larger adversarial loss) consistently harm the model performance across these evaluations, which shows the spirit of loss-maximization in Re-SFAT and AFL is contrary to the correct way. In contrast, emphasizing the client with smallest adversarial loss indeed improve the model performance in terms

of both natural and robust accuracies. It confirms the rationality of SFAT to alleviating intensified heterogeneity by relaxing the inner-maximization of adversarial generation during aggregation.

Note that the generalization focus by AFL is under the standard federated learning instead of federated adversarial training, and the empirical results also confirm the loss-maximization actually trigger the intensified heterogeneity and lead to lower accuracies (in Table 10). Since it is out of the scope of this work on handling the intensified heterogeneity, we leave it to our future work.

With respect to the experiments with standard training (in the left of Figure 4), there is no inner-maximization in Eq.(3). Thus, there is no intensified heterogeneity to handle but only positive training signals from the standard training. In this case, adding the inequality with slack in Eq.(3) only induces the extra objective bias to the standard federated learning. Correspondingly, as shown in the left-most panel of Figure 4, such an operation instead degenerates the model performance.

**Discussion about the loss and the intensified heterogeneity.** It is the learning dynamic of Eq. 3 that upweights the client model with the smaller losses and downweights the client model with the larger losses to slack the overall objective. Possibly, the optimization bias (or termed as client drift more rigorously) may result in a smaller loss and sometimes the smaller loss may not absolutely indicate the smaller optimization bias. We have verified that when the opposite does not hold and the larger loss is preferred in the selection in the middle panel of Figure 4 (see Re-SFAT v.s. SFAT) and the above Table 10 (see Re-SFAT v.s. SFAT). The Re-SFAT is actually construct an upper bound of the FAT objective, which is different from SFAT that relax to the lower bound of the FAT objective. The results empirically verify the failure of this case that prefers the larger loss. Here we discuss the connection between the slack mechanism and the intensified heterogeneity (or roughly call intensified optimization bias). To alleviate the intensified heterogeneity, our slack mechanism naturally relaxes the overall objective and constructs a mediating function that asymptotically approaches the original goal and reducing the negative impact of the intensified heterogeneity on the training. Our reasonable analysis and comprehensive evidence from both theoretical (e.g., Theorems 4.2 and 4.3) and empirical (e.g., Figures 3, 10, and 11) views have demonstrated its rationality. In Tables 3 and 19, the multiple experimental results with random non-iid data distribution as well as the real-world datasets demonstrates the empirical superiority of our proposed SFAT over the original FAT.

### E.4    SFAT CORRESPONDING TO CLIENT DRIFT

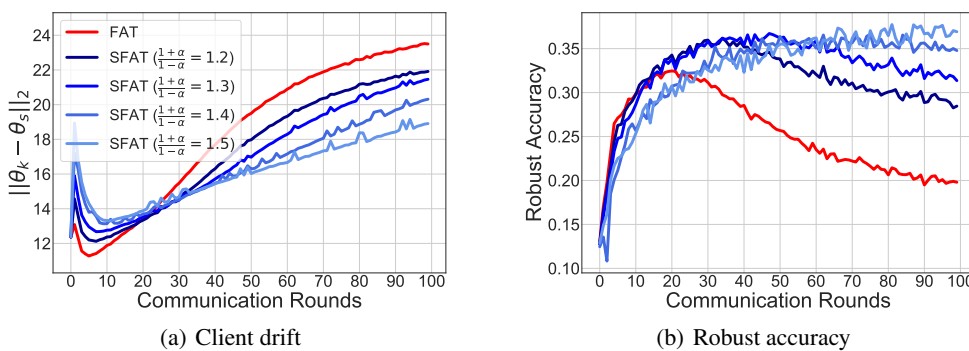

(a) Client drift                    (b) Robust accuracy

Figure 10: The client drift corresponding to the robust accuracy of FAT and our SFAT during training.

In this part, we present more results about the client drift during training using FAT and our SFAT than Figure 3. The corresponding curve during training can be more clearly to show how SFAT alleviate the intensified heterogeneity and achieve the higher robust accuracy.

We conduct more experiments about the client drift of FAT and our SFAT on *CIFAR-10* (Non-IID) in Figure 10. As the $\alpha$ increasing, our SFAT can further alleviate the intensified heterogeneity (having smaller client drift value as shown in Figure 10(a)) of FAT at the later stage which corresponds to the robustness deterioration (exhibiting less obvious deterioration while achieving higher robust accuracy as shown in Figure 10(b)). Unlike FAT that hindered by the intensified heterogeneity, our SFAT can improve the robust accuracy by combating the issue of intensified heterogeneity.

### E.5 ADDITIONAL DISCUSSION ON THE $\alpha$-SLACK MECHANISM

Here, we first empirically verify the difference between FedSoftBetter and our SFAT, we conduct the experiments on CIFAR-10 and SVHN to compare their performance in the following table. The results show that its empirical performance is slightly better than FedAvg but not better than SFAT.

Table 11: Comparison of different methods on CIFAR-10 and SVHN datasets.

| Dataset | Methods | Natural | FGSM | PGD-20 | CW$_\infty$ |
|---|---|---|---|---|---|
| CIFAR-10 | FAT | 57.45% | 39.44% | 32.58% | 30.52% |
| | FedSoftBetter | 58.86% | 40.23% | 32.78% | 30.76% |
| | **SFAT** | **63.44%** | **45.13%** | **37.17%** | **33.99%** |
| SVHN | FAT | 91.24% | 87.95% | 68.87% | 67.39% |
| | FedSoftBetter | **91.64%** | **88.53%** | 69.07% | 67.83% |
| | **SFAT** | 91.25% | 88.28% | **71.72%** | **69.79%** |

In the following, we present the experiments that progressively anneal the coefficient to vanilla federated adversarial training during the training and compare with SFAT on CIFAR-10 and SVHN. According to the results in Table 12, gradually annealing-$\alpha$ SFAT achieves slightly lower robust accuracy than the constant-$\alpha$ SFAT and sometimes may introduces the large drop of natural accuracy (e.g., on CIFAR-10). It indicates that keeping a consistent $\alpha$ for the $\alpha$-slack maybe a better choice. On the other hand, it also confirms the intuition that using a gradually decreased alpha is empirically contrary to the observation of robustness deterioration (or the heterogeneity exacerbation as shown in Figure 10) at the later stage of training.

Table 12: Comparison on CIFAR-10 and SVHN datasets using different $\alpha$ schedule.

| Dataset | $\alpha$-slack | Methods | Natural | FGSM | PGD-20 | CW$_\infty$ |
|---|---|---|---|---|---|---|
| CIFAR-10 | - | FAT | 57.45% | 39.44% | 32.58% | 30.52% |
| | $\frac{1+\alpha}{1-\alpha} : 1.4$ | **SFAT** | **63.44%** | **45.13%** | **37.17%** | **33.99%** |
| | $\frac{1+\alpha}{1-\alpha} : 1.4 \rightarrow 1.0$ | **SFAT** | 60.63% | 43.53% | 36.23% | 33.22% |
| SVHN | - | FAT | 91.24% | 87.95% | 68.87% | 67.39% |
| | $\frac{1+\alpha}{1-\alpha} : 1.2$ | **SFAT** | 91.25% | 88.28% | **71.72%** | **69.79%** |
| | $\frac{1+\alpha}{1-\alpha} : 1.2 \rightarrow 1.0$ | **SFAT** | **91.68%** | **88.55%** | 71.44% | 69.76% |

### E.6 SFAT CORRESPONDING TO GRADIENT VARIANCE

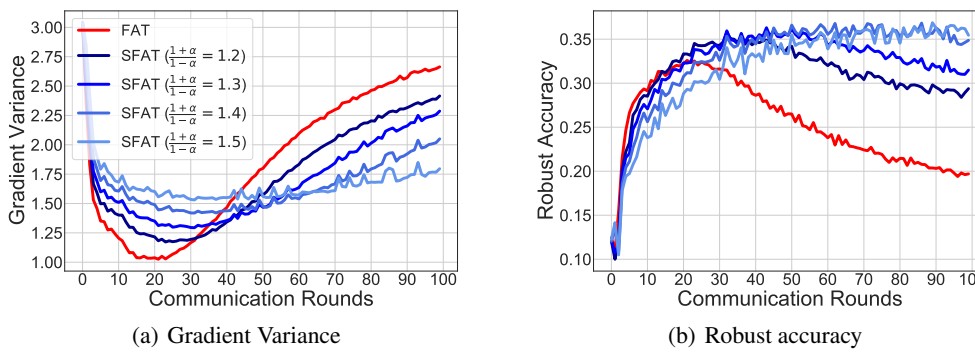

(a) Gradient Variance

(b) Robust accuracy

Figure 11: The gradient variance corresponding to the robust accuracy of FAT and our SFAT.

To further factorize how the inner maximization affects the intensified heterogeneity of the local data, we add the experiments about the variance of the client's gradients (calculated by parameters difference). We find the similar trend with the empirical results about client drift in Figure 11, i.e., SFAT prevents exacerbated gradients variance in FAT, and results in better performance on robustness.

### E.7 EMPIRICAL VERIFICATION ABOUT OUR THEORETICAL ANALYSIS.

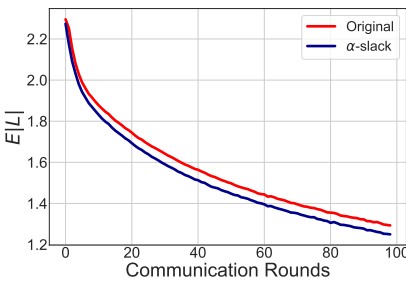 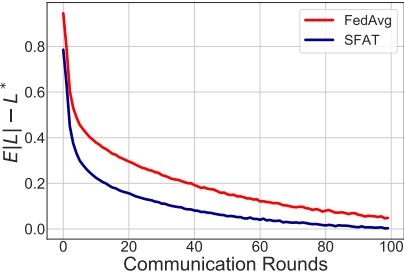

Figure 12: Empirical verification of our theoretical analyze. Left panel: Empirical estimation about $\mathbb{E}|\cdot|$ in Theorem 4.2 of the original and our slacked objective using CIFAR10 dataset. Right panel: Empirical estimation about $\mathbb{E}|\mathcal{L}_{\text{SFAT}}| - \mathcal{L}^*$ in Theorem 4.3 of FAT and our SFAT using SVHN dataset. The overall results confirm the benefit on convergence from our proposed $\alpha$-slack mechanism.

Here, we provide the empirical verification about the theoretical claim in Theorem 4.2 and Theorem 4.3 via tracing the training loss on CIFAR-10 and SVHN datasets. In Figure 12, we present the empirical estimation of the RHS of Eq. (2) and Eq. (4) via tracing the training loss. The results show that our our $\alpha$-slack mechanism can achieve a faster convergence compared with the original objective for the convergence of adversarial training in Theorem 4.2 and the federated case in Theorem 4.3.

### E.8 EXPERIMENTS ABOUT THE ORTHOGONAL EFFECTS OF SFAT ON CLIENT DIRFT.

In this part, we start with the discussion about the orthogonal effects of SFAT from the conceptual perspective, and then present the orthogonal effects of SFAT on combating the intensified heterogeneity via tracing the client drift from the experimental perspective.

From the problem view, our slack mechanism is targeted for the intensified heterogeneity. However, similar to other techniques in federated learning literature, FedProx (Li et al., 2018) is designed for the original heterogeneous data instead of considering the intensified process. The two problems are discussed in our Appendix A. Although intensified heterogeneity and the ordinary heterogeneity (i.e., without the inner-maximization) both induce the client drift, the effects and extent are different. To be specific, the intensified heterogeneity will result in more diverged models and exacerbate the difference among client models compared with the ordinary heterogeneity. This intensification process is not considered in FedProx or other related work (McMahan et al., 2017; Karimireddy et al., 2020), and SFAT is orthogonal to them further incorporates a slack mechanism to avoid its influence.

Table 13: Client drift w.r.t Epochs on CIFAR-10 in Figure 3.

| Setting | | Client drift ↓ | | | | | | | | | |
|---|---|---|---|---|---|---|---|---|---|---|---|
| Method / Epoch | | 10 | 20 | 30 | 40 | 50 | 60 | 70 | 80 | 90 | 100 |
| FedAvg | FAT | **11.56** | 13.10 | 15.04 | 17.30 | 19.22 | 20.66 | 21.72 | 22.47 | 23.02 | 23.49 |
| | **SFAT** | 12.21 | **13.08** | **14.37** | **15.95** | **17.68** | **19.04** | **20.16** | **20.92** | **21.50** | **21.86** |
| FedProx | FAT | **4.71** | **4.58** | 5.98 | 6.54 | 7.83 | 8.26 | 9.12 | 10.23 | 11.71 | 12.61 |
| | **SFAT** | 5.89 | 5.27 | **5.67** | **6.02** | **6.87** | **7.34** | **7.98** | **8.77** | **9.65** | **10.81** |

From the experimental view, we also provide the comparison between SFAT and FedProx (actually they are orthogonal and combinable). We show the results of client drift w.r.t different methods in the Tables 13 and 14. According to the results, FAT under the backbone FedProx can indeed catch up with our SFAT on basis of FedAvg since the FedProx is designed for reducing the client drift. However, when adopting FedProx as the backbone of SFAT, the intensified client drift can be further reduced, which verified the orthogonal effects (as stated in our Appendix A) on reducing intensified heterogeneity in the critical issue of federated adversarial training. Note that, we are not focus on the relationship of normal heterogeneity (Li et al., 2018) (e.g., the value of client drift) with the robust

Table 14: Client drift w.r.t Epochs on CIFAR-10 in the setting of unequal splits with 5 clients.

| Setting | | Client drift ↓ | | | | | | | | | |
|---------|------|-------|-------|-------|-------|-------|-------|-------|-------|-------|-------|
| Method / Epoch | | 10 | 20 | 30 | 40 | 50 | 60 | 70 | 80 | 90 | 100 |
| FedAvg | FAT | **13.96** | 16.91 | 19.89 | 22.52 | 24.40 | 25.75 | 26.76 | 27.64 | 28.13 | 28.53 |
| | SFAT | 14.15 | **16.71** | **19.37** | **21.68** | **23.33** | **24.24** | **25.39** | **26.11** | **26.75** | **26.93** |
| FedProx | FAT | **5.36** | **5.89** | 6.82 | 7.96 | 9.36 | 10.89 | 12.52 | 14.03 | 15.41 | 16.47 |
| | SFAT | 6.28 | 6.27 | **6.59** | **7.10** | **7.68** | **8.10** | **9.06** | **9.80** | **10.60** | **11.14** |

performance in federated adversarial training. Instead, our proposed SFAT actually focus on the robust deterioration caused by the intensified heterogeneity (e.g., the increasing trend of client drift). It is orthogonal and compatible to those previous federated optimization methods.

Table 15: Comparison of SFAT and FAT using FedProx with different paramter $\mu$.

| FedProx: $\mu$ | Methods | Natural | PGD-20 | CW$_\infty$ |
|----------------|---------|---------|--------|-------------|
| 0.01 | FAT | 90.92% | 68.44% | 67.18% |
| | **SFAT** | **91.25%** | **71.54%** | **69.53%** |
| 0.05 | FAT | 90.25% | 68.07% | 66.88% |
| | **SFAT** | **91.37%** | **70.58%** | **68.93%** |
| 0.1 | FAT | 89.98% | 67.15% | 65.94% |
| | **SFAT** | **90.95%** | **71.89%** | **69.98%** |

Except the previous results, we further strength the hyper-parameter $\mu$ of the proposed proximal term i.e., $\frac{\mu}{2}||w - w^t||_2$ in FedProx to verify the improvement of using our SFAT on SVHN dataset. We summarize the results in Table 15. The results show that increasing the $\mu$ from 0.01 (adopted in our experiments and following the recommendation of FedProx) to 0.1, the robust performance even worse while our SFAT can still reach the better performance. The reason may be that too large $\mu$ also has the potentially negative influence on the convergence of the training by forcing the updates to be close to the starting point, which has been discussed in previous literature (Karimireddy et al., 2020).

### E.9 EXPERIMENTS WITH MORE CLIENTS

Table 16: Comparison about FAT with SFAT on Non-IID data partition with different client numbers

| Setting | | | Non-IID | | | | | | | | |
|---------|---|------|---------|--------|------|---------|--------|------|-----------|--------|------|
| | | | CIFAR-10 | | | SVHN | | | CIFAR-100 | | |
| Client Number / Method | | | Natural | PGD-20 | CW$_\infty$ | Natural | PGD-20 | CW$_\infty$ | Natural | PGD-20 | CW$_\infty$ |
| FedAvg | 10 | FAT | 56.62% | 31.24% | 29.82% | 91.42% | 69.65% | 68.52% | 33.27% | 16.81% | 14.12% |
| | | SFAT | **56.67%** | **33.31%** | **31.58%** | **91.84%** | **72.59%** | **70.71%** | **34.17%** | **17.66%** | **14.25%** |
| | 20 | FAT | 60.55% | 32.67% | 31.07% | 92.14% | 70.32% | 69.48% | 31.49% | 15.35% | 13.18% |
| | | SFAT | **62.24%** | **35.66%** | **33.21%** | **92.75%** | **72.06%** | **71.14%** | **34.04%** | **16.05%** | **13.70%** |
| | 25 | FAT | 58.97% | 32.98% | 31.14% | 92.32% | 70.54% | 69.84% | 32.64% | 15.82% | 13.23% |
| | | SFAT | **62.73%** | **35.75%** | **33.16%** | **92.33%** | **71.99%** | **71.06%** | **34.19%** | **16.37%** | **13.63%** |
| | 50 | FAT | 56.74% | 32.91% | 30.50% | 91.97% | 70.84% | 69.42% | 34.46% | 15.97% | 13.59% |
| | | SFAT | **57.21%** | **34.35%** | **31.75%** | **91.99%** | **71.87%** | **70.74%** | **34.82%** | **16.34%** | **13.93%** |

In this part, we verify SFAT using more clients under the Non-IID data partition with the three benchmarked datasets, i.e., *CIFAR-10*, *SVHN* and *CIFAR-100*.

In Table 16, we change the client number from 10 to 50 to investigate the scalability of our SFAT. For each client setting, we conduct FAT and SFAT to compare their performance on both natural test data and adversarial test data. In the experiments, we set $\widehat{K} = K/5$ and $\alpha = 1/11$ for our SFAT and keep the other basic setups as the same with previous experiments. We can find that the results further confirm the effectiveness of SFAT on improving both natural and robust performance when training with different client numbers.

## E.10 EXPERIMENTS ON THE UNEQUAL DATA SPLITS.

Table 17: Performance on the setting with unequal data splits among clients.

| Setting | | | | | Non-IID | | | |
|---|---|---|---|---|---|---|---|---|
| Dataset | Client | Sample | Opt. | Method | Natural | FGSM | PGD-20 | CW$_\infty$ |
| CIFAR-10 | 5 | 6000-13000 | FedAvg | FAT | 59.98% | 40.57% | 31.50% | 29.57% |
| | | | | SFAT | **61.70%** | **42.81%** | **33.87%** | **30.99%** |
| | | | FedProx | FAT | 60.36% | 40.36% | 31.90% | 29.00% |
| | | | | SFAT | **60.65%** | **42.38%** | **35.16%** | **30.93%** |
| | 10 | 1000-8000 | FedAvg | FAT | 61.67% | 42.69% | 33.17% | 30.58% |
| | | | | SFAT | **62.57%** | **44.85%** | **36.42%** | **32.65%** |
| | | | FedProx | FAT | 60.24% | 41.25% | 33.21% | 30.98% |
| | | | | SFAT | **60.78%** | **43.73%** | **36.76%** | **32.44%** |
| Dataset | Client | Sample | Opt. | Method | Natural | FGSM | PGD-20 | CW$_\infty$ |
| SVHN | 5 | 5860-26370 | FedAvg | FAT | 89.42% | 85.93% | 68.35% | 67.03% |
| | | | | SFAT | **90.57%** | **87.53%** | **70.56%** | **68.67%** |
| | | | FedProx | FAT | 90.15% | 86.59% | 68.02% | 66.22% |
| | | | | SFAT | **90.55%** | **87.45%** | **71.19%** | **69.00%** |
| | 10 | 1465-13185 | FedAvg | FAT | **91.84%** | 88.80% | 70.66% | 68.90% |
| | | | | SFAT | 91.55% | **88.91%** | **72.29%** | **70.30%** |
| | | | FedProx | FAT | 90.95% | 87.77% | 69.80% | 68.20% |
| | | | | SFAT | **91.61%** | **88.83%** | **72.45%** | **70.29%** |

Table 18: Performance on the setting with severe unequal splits among clients on SVHN dataset.

| Setting | | | | | Non-IID | | | |
|---|---|---|---|---|---|---|---|---|
| Dataset | Client | Sample | Opt. | Method | Natural | FGSM | PGD-20 | CW$_\infty$ |
| SVHN | 10 | 1465-13185 | FedAvg | FAT | **91.84%** | 88.80% | 70.66% | 68.90% |
| | | | | SFAT | 91.55% | **88.91%** | **72.29%** | **70.30%** |
| | | | FedProx | FAT | 90.95% | 87.77% | 69.80% | 68.20% |
| | | | | SFAT | **91.61%** | **88.83%** | **72.45%** | **70.29%** |
| | 10 | 50-16700 | FedAvg | FAT | 93.14% | 90.23% | 72.09% | 71.01% |
| | | | | SFAT | **92.78%** | **90.07%** | **73.85%** | **72.08%** |
| | | | FedProx | FAT | 93.06% | 90.37% | 72.03% | 70.90% |
| | | | | SFAT | **93.14%** | **90.51%** | **73.87%** | **72.35%** |

To complete our experimental verification on those unequal data splits, we conduct the experiments on *CIFAR-10* and *SVHN* datasets with different client numbers. We summarize the results in Table 17. In addition, we also add Table 18 to explore the more severe unequal splits in different clients.

In such setups, the sample numbers of different client are also a critical factor to the optimization. More samples the client has, larger weights the local model has in the aggregation phase. In comparison with our $\alpha$, it follows the support of the statistical sample proportion while our $\alpha$ utilizes the clues of the local loss related to the adversarial training. According to the results, we can see that SFAT is approximately orthogonal to the sample number effect and still more effective than FAT.

From the algorithm level, our algorithm (refer to Algorithm 1) and the framework (refer to Figure 6) has taken the numbers of local data into consideration. To be specific, when we conduct the slack selection, all the adversarial training loss are normalized by the local data number.

## E.11 EXPERIMENTS ON THE REAL-WORLD SCENARIOS

To verify the effectiveness of our SFAT in more practical situation, we conduct the experiments using a real-world dataset *CelebA* in the benchmark of federated learning, i.e., LEAF (Caldas et al., 2018),

Table 19: Performance on Non-IID settings using the real-world dataset, i.e., *CelebA*

| Setting | | Non-IID | | | |
|---|---|---|---|---|---|
| CelebA | | Natural | FGSM | PGD-20 | $CW_\infty$ |
| FedAvg | FAT | 57.62% | 42.20% | 22.20% | 21.67% |
| | **SFAT** | **58.50%** | **43.44%** | **24.14%** | **23.52%** |
| FedProx | FAT | 57.70% | 41.85% | 22.29% | 21.50% |
| | **SFAT** | **58.50%** | **43.08%** | **24.14%** | **23.61%** |

with hundreds (455) of clients in Table 19. We follow the most settings in LEAF to perform our experiments using different federated optimization methods, and we set $\widehat{K} = 2 * K/5$ and $\alpha = 1/6$ for all the experiments of our SFAT.

In Table 19, we confirm the effectiveness of our SFAT using a real-world large-scale dataset *CelebA* with 455 clients. Except Scaffold that fails to converge in FAT and thus is not reported, our SFAT again gains significantly better natural and robust accuracies than FAT under FedAvg and FedProx.

Table 20: Test accuracy (%) on *SVHN* with different participation ratio.

| Setting | | Non-IID | | | | |
|---|---|---|---|---|---|---|
| Accuracy / Participation Ratio | | 0.2 | 0.4 | 0.6 | 0.8 | 1.0 |
| Natural | FAT | **91.93%** | 91.39% | 92.19% | 92.31% | 92.14% |
| | **SFAT** | 91.61% | **92.51%** | **92.30%** | **92.53%** | **92.75%** |
| PGD-20 | FAT | 69.63% | 69.97% | 70.25% | 70.24% | 70.32% |
| | **SFAT** | **72.37%** | **73.01%** | **72.85%** | **72.59%** | **72.06%** |

In addition, we also consider the practical situation where only a subset of clients participates in each round. Following the same settings as previous section, we add the experiments on *SVHN* dataset in Table 20 with 20 clients. The results show that the lower participation ratio leads to lower natural and PGD-20 accuracy while our SFAT can consistently outperform FAT on the robustness.

### E.12 OVERALL RESULTS ON BOTH NON-IID AND IID SETTINGS

Here we provide the overall results for comparison on both Non-IID and IID settings in Table 21.

For the Non-IID data, our SFAT gain consistently improvement across the various of evaluation metrics and datasets. For the IID data, our method acquires a similar improvement on the robust accuracy without the deterioration of the natural accuracy. The reason might be that even the data is IID, adversarial training can still drive the independently-initialized overparameterized network (Allen-Zhu & Li, 2020) on each client side towards at the robust overfitting of different directions, yielding the model heterogeneity. Thus, the proper slack to the inner-maximization makes adversarial training more compatible with federated learning. Another interesting observation is that Federated Adversarial Training shows better performance than centralized adversarial training in the IID setting. This gain could be from the distributed training paradigm that helps adversarial training converge to the more robust optimum by the divide-and-conquer mechanism. This might enlighten the more explore in adversarial training to improve the robustness via federated learning.

Table 21: Performance on three benchmark datasets under different federated optimization methods (Non-IID & IID).

| Setting | | Non-IID | | | | | IID | | | | |
|---|---|---|---|---|---|---|---|---|---|---|---|
| CIFAR-10 | | Natural | FGSM | PGD-20 | $CW_\infty$ | AA | Natural | FGSM | PGD-20 | $CW_\infty$ | AA |
| Centralized AT | | - | - | - | - | - | 66.47% | 47.68% | 38.18% | 37.04% | 34.48% |
| FedAvg | FAT | 57.45% | 39.44% | 32.58% | 30.52% | 29.20% | **69.35%** | 48.45% | 37.43% | 35.72% | 33.96% |
| | **SFAT** | **63.44%** | **45.13%** | **37.17%** | **33.99%** | **32.36%** | 67.43% | **50.33%** | **42.78%** | **37.91%** | **36.20%** |
| FedProx | FAT | 60.44% | 41.59% | 33.84% | 31.29% | 30.02% | 66.91% | 46.70% | 37.14% | 34.54% | 32.68% |
| | **SFAT** | **62.51%** | **44.29%** | **36.75%** | **33.82%** | **31,98%** | **68.31%** | **48.40%** | **42.41%** | **37.25%** | **35.97%** |
| Scaffold | FAT | 62.81% | 43.61% | 34.13% | 32.53% | 30.95% | 68.27% | 49.25% | 39.33% | 37.31% | 35.30% |
| | **SFAT** | **64.12%** | **46.05%** | **37.35%** | **34.78%** | **33.32%** | **71.36%** | **50.42%** | **43.83%** | **39.12%** | **35.47%** |
| CIFAR-100 | | Natural | FGSM | PGD-20 | $CW_\infty$ | AA | Natural | FGSM | PGD-20 | $CW_\infty$ | AA |
| Centralized AT | | - | - | - | - | - | 35.81% | 23.09% | 18.64% | 16.48% | 15.42% |
| FedAvg | FAT | 35.19% | 20.20% | 15.60% | 13.26% | 12.22% | 32.65% | 20.44% | 16.47% | 14.10% | 12.99% |
| | **SFAT** | **36.18%** | **20.70%** | **16.40%** | **13.55%** | **12.42%** | **38.36%** | **21.86%** | **17.10%** | **14.36%** | **13.42%** |
| FedProx | FAT | 32.36% | 19.22% | 15.37% | 12.91% | 12.05% | 34.78% | 20.71% | 16.37% | 14.28% | 13.09% |
| | **SFAT** | **35.11%** | **20.62%** | **16.19%** | **13.47%** | **12.63%** | **37.58%** | **21.74%** | **17.03%** | **14.46%** | **13.50%** |
| Scaffold | FAT | 39.96% | 24.26% | 19.41% | 16.60% | 15.37% | 43.80% | 26.25% | 20.76% | 18.39% | 17.20% |
| | **SFAT** | **44.08%** | **24.38%** | **20.29%** | **16.79%** | **15.90%** | **44.36%** | **28.65%** | **23.14%** | **20.11%** | **18.39%** |
| SVHN | | Natural | FGSM | PGD-20 | $CW_\infty$ | AA | Natural | FGSM | PGD-20 | $CW_\infty$ | AA |
| Centralized AT | | - | - | - | - | - | 92.39% | 89.75% | 72.73% | 72.31% | 70.93% |
| FedAvg | FAT | 91.24% | 87.95% | 68.87% | 67.89% | 66.54% | **93.52%** | **90.68%** | 72.24% | 71.22% | 70.08% |
| | **SFAT** | **91.25%** | **88.28%** | **71.72%** | **69.79%** | **68.62%** | 92.75% | 90.06% | **74.37%** | **72.34%** | **71.27%** |
| FedProx | FAT | 90.92% | 87.50% | 68.44% | 67.18% | 65.94% | 93.54% | 90.66% | 72.53% | 71.42% | 70.21% |
| | **SFAT** | **91.25%** | **88.15%** | **71.54%** | **69.53%** | **68.47%** | **93.59%** | **90.80%** | **74.66%** | **72.67%** | **71.48%** |
| Scaffold | FAT | 89.95% | 87.23% | 68.66% | 67.23% | 66.65% | 93.80% | 91.00% | 73.26% | 72.05% | 70.80% |
| | **SFAT** | **90.20%** | **87.81%** | **71.39%** | **68.81%** | **67.88%** | **93.92%** | **91.28%** | **75.96%** | **74.05%** | **72.88%** |

# F   FURTHER DISCUSSION

Adversarial robustness is an important topic in the centralized machine learning. The adversarial training are confirmed to be one of the most effective empirical defenses against the adversarial attack, which is critical especially for those safety-critical areas like medicine and finance. In federated settings, how to train an adversarially robust model is a challenging but practical task for the increasing concern about data privacy. In this work, we observe and explore to combat the intensified heterogeneity in federated adversarial training. Different from the conventional FAT adopted by previous works, we propose a new learning framework, i.e., SFAT, which relaxes the exacerbated heterogeneous effect and is compatible with the various adversarial training (Madry et al., 2018; Zhang et al., 2019) and federated optimization methods (Li et al., 2018; Karimireddy et al., 2020).

Although we take a step forward in FAT, it is not the end of this direction since there are still many problems to be addressed to further enhance the practicality of federated adversarial training.

From the perspective of adversarial robustness, adversarial attacks can be very complex, especially in a decentralized environment, while the adversarial robustness discussed in this paper mainly focuses on common adversarial attacks (e.g., $L_\infty$-bounded attack) (Goodfellow et al., 2015). More practical situation which contains different kind of adversarial attack (e.g., mixed types of attack with $L_\infty$-bounded attack, Spatially transformed attack (Xiao et al., 2018)), even only considering the inference phase, may also happened since there are different clients may meet different threaten (Kairouz et al., 2019; Yao et al., 2022). Besides, the federated adversarial training that requires multiple local runs (Madry et al., 2018; Zhang et al., 2019) also introduces the extra computation to the low-capacity devices, which is computational bottleneck and requires some lightweight techniques. Except for the empirical defense strategy focused by our work, the certifiable robustness (Cohen et al., 2019; Zizzo et al., 2021; Alfarra et al., 2022) which can give the theoretical guarantees is also important.

From the perspective of federated learning, our SFAT shares the similar spirit of the conventional federated adversarial training. The first part of challenges comes from the distributed learning

paradigm (McMahan et al., 2017; Kairouz et al., 2019; Li et al., 2020) of federated setting, which brings the hardware constraint that considers the computational capacity of local clients and communication cost between clients and server (Hong et al., 2021). Once these conditions do not satisfy, both FAT and SFAT would not work well. The second may from the algorithm design for some special training or inference issues, like dealing with heterogeneous data (Li et al., 2018; Zhao et al., 2018), class-imbalance data or even some out-of-distribution data at inference time. On the other hand, the decentralized structure of the learning paradigm also introduces various issues on information transferring for server and clients. The current federated adversarial training still needs large improvement considering the practical cases that may happened in the federated setting. For the intensified heterogeneity, it can also be recognized as an dynamical heterogeneous issue existing in federated learning, which may result from the special learning algorithm adapted in the distributed framework or other data manipulation scenarios. More robust issues, like robust distillation (Goldblum et al., 2020; Zhu et al., 2022), train-test distribution shift (Jiang & Lin, 2023), out-of-distribution detection (Yu et al., 2023), under the federated framework can be further explored in the future.

