# OpenReview forum: "Combating Exacerbated Heterogeneity for Robust Models in Federated Learning"
_ICLR.cc/2023/Conference — ICLR 2023 poster_

### Official Review · Reviewer_YWt9 · 2022-10-24

**Confidence:** 5
**Correctness:** 4
**Technical Novelty And Significance:** 4
**Empirical Novelty And Significance:** Not applicable
**Recommendation:** 8

**Clarity, Quality, Novelty And Reproducibility:**

The authors really conduct an impressive work and give a range of experiments and the corresponding theoretical analysis to demonstrate the advantages of the relaxing mechanism.

**Strength And Weaknesses:**

In a nutshell, the authors investigate an interesting problem motivated by the degeneration observation, and the proposed method is solid with sufficient evidences. However, there are still some concerns regarding the submission as follows,
(1)	Although the combination of two paradigms lead to the degeneration in performance, the reason that induces this phenomenon needs to be carefully discussed, since all toy studies even including the analysis on real-world dataset are from the perspective of the observation, which cannot attribute to the real cause. For example, it is not absolute in Figure 2 where the generated adversarial examples move in the clockwise direction, instead of the converse direction. I advise the authors should properly express the conjecture in the motivation part.
(2)	Equation (1) introduces a lower bound to the original objective, which is different from the general way that constructs an upper bound to minimize. Therefore, it is not necessary that minimizing a lower bound can lead to the minimization of the original objective. The authors should discuss more here and give some empirical validation if it is possible.
(3)	The background part gives some basics of federated learning and adversarial training, while the review of current works is insufficient. I advise the reviewer should follow the conventional style and separate this part into related work and preliminary, respectively for literature review and basics.
(4)	In tables 1-4, the natural accuracy and the robustness performance in the centralized counterpart can be validated as references, and the gap in comparison to the achieved performance in federated learning framework should be reviewed so that the readers can deeply understand the limitation in this topic.
(5)	Again, I think the explanation about the problem in the combination of federated learning and adversarial training should be improved. Possibly, if the authors can find some supporting examples from the experiments as the visualization or some statistical analysis can make the claim more convincing. I advise the reviewers to trace the classification difference between the baseline and the SFAT-enhanced counterpart and analyze the common characteristics of these non-overlapping examples.
(6)	The discussion about the systematic challenges in federated adversarial training is limited, yielding that it is hard to evaluate the value of this narrow direction. More analysis about the practical requirement in the real-world applications should be given both in the introduction and related work as well as the confusion remark.


**Summary Of The Paper:**

This paper studies the adversarial robustness of federated learning, which is an important problem when training the model on the client side. Specially, the authors consider the incompatible dilemma between federated learning and adversarial training, and introduce a mediating slack mechanism to better bridge two paradigms. Both theoretical analysis and empirical study demonstrate the effectiveness of the proposed methods.

**Summary Of The Review:**

My main worry is not about the observation and the proposed method, but about the motivation description and analysis is lack of some sufficient justification. Please carefully consider above concerns and improve the submission, and I can consider to raise the score if they are well solved.

---

> ### Author Response · Authors · 2022-11-12
> **Response to Reviewer YWt9 [2/2]**
>
> > **Q5:** Again, I think the explanation about the problem in the combination of federated learning and adversarial training should be improved. Possibly, if the authors can find some supporting examples from the experiments as the visualization or some statistical analysis can make the claim more convincing. I advise the reviewers to trace the classification difference between the baseline and the SFAT-enhanced counterpart and analyze the common characteristics of these non-overlapping examples.
>
> **A5:** Thanks for your suggestion. We will replace the left panel of Figure 2 with Figure 6(a) that is more empirically verified, and revise the corresponding explanation with the proper description. Following the suggestion, we conducted extra experiments to investigate the classification difference between the baseline and the SFAT. However, it is hard to draw and understand the characteristics from the raw data level regarding those non-overlapped examples. To better support our claim, we compared the robust deterioration in FAT with different adversarial training strengths (using the same robust evaluation strength) and summarized the results in the following table. The results show that more stronger adversarial generation (i.e., the larger inner-maximization) leads to more severe robustness deterioration (indicated by the large accuracy gap between the best and last stage). It confirms the rationality of our derived lower bound by $\alpha$-slack mechanism to pursue the adversarial robustness and alleviate the heterogeneity exacerbation.
>
> **Table 1.** Robust deterioration w.r.t adversarial training strength in FAT.
>
> |          |               |       | Performance gap between best and last epoch |            |
> | :------: | :-----------: | :---: | :-----------------------------------------: | :--------: |
> | Dataset  | Adv. Strength | 2/255 |                    4/255                    |   8/255    |
> | CIFAR-10 |      FAT      | 6.33% |                   11.16%                    | **13.06%** |
> | Dataset  | Adv. Strength | 1/255 |                    2/255                    |   4/255    |
> |   SVHN   |      FAT      | 4.65% |                    5.63%                    | **6.43%**  |
>
>
> > **Q6:** The discussion about the systematic challenges in federated adversarial training is limited, yielding that it is hard to evaluate the value of this narrow direction. More analysis about the practical requirement in the real-world applications should be given both in the introduction and related work as well as the confusion remark.
>
> **A6:** Thanks for your suggestions. We provide the part of the discussion about the challenges of federated adversarial training in our Appendix F.
>
> Adversarial training is important to enhance the robustness of machine learning models against adversarial attacks, especially for some safety-critical areas like medicine and finance. In these areas, there is also the increasing concern about data privacy. Hence, it raises a significant and practical challenge in training the adversarially robust model in federated settings. As for the challenges in federated adversarial training, it comes from various aspects. Most generally, we provide the discussion from two perspectives. The first is the distributed learning paradigm of the federated setting, which brings the hardware constraint that considers the computational capacity of local clients and communication cost between clients and the server. In this perspective, adversarial training introduces extra computation costs to those local clients, which is a computational bottleneck and requires lightweight techniques to improve it. The second is from the algorithm design for some special training or inference issues in both the federated setting and adversarial training, like dealing with heterogeneous data, class-imbalance data, or even some out-of-distribution data at inference time. There is still a certain gap between the conventional federated adversarial training framework and the more practical real-world scenarios.
>
> We will add the detailed discussion as well as more analysis of the practical requirements in real-world applications in the introduction and related work parts.

---

> ### Author Response · Authors · 2022-11-12
> **Response to Reviewer YWt9 [1/2]**
>
> Thank you for your time devoted to reviewing this paper and your constructive suggestions. Here are our detailed replies to your questions.
>
> > **Q1:** Although the combination of two paradigms lead to the degeneration in performance, the reason that induces this phenomenon needs to be carefully discussed, since all toy studies even including the analysis on real-world dataset are from the perspective of the observation, which cannot attribute to the real cause. For example, it is not absolute in Figure 2 where the generated adversarial examples move in the clockwise direction, instead of the converse direction. I advise the authors should properly express the conjecture in the motivation part.
>
> **A1:** Thanks for your suggestion. We will revise the introduction and motivation parts to properly refine the description about the conjecture on the robust degeneration. Besides, for the illustration part, we would also substitute the left panel of Figure 2 with Figure 6(a) as real-world empirical verification support. It will be revised in our updated submission.
>
>
> > **Q2:** Equation (1) introduces a lower bound to the original objective, which is different from the general way that constructs an upper bound to minimize. Therefore, it is not necessary that minimizing a lower bound can lead to the minimization of the original objective. The authors should discuss more here and give some empirical validation if it is possible.
>
> **A2:** We would like to clarify that our intuition is to relax the original objective to alleviate the heterogeneity exacerbation. Although it is true that minimizing the lower bound does not always mean the minimization of the original objective, the similar optimization strategies also occur in the optimization of discrete variables like Gumbel-SoftMax [1], which is indefinite in terms of the bound relationship. However, it is really effective in the empirical results. We appreciate the advice of the reviewer and will add more discussion about the corresponding parts around Eq.(1).  Empirically, we present the validation in Figures 10, 11, and 12 to show that our SFAT can alleviate the heterogeneity exacerbation and achieve the better robustness by relaxing the objective to a lower bound.
>
> [1] Jang E, Gu S, Poole B. Categorical reparameterization with gumbel-softmax[J]. ICLR, 2017.
>
> > **Q3:** The background part gives some basics of federated learning and adversarial training, while the review of current works is insufficient. I advise the reviewer should follow the conventional style and separate this part into related work and preliminary, respectively for literature review and basics.
>
> **A3:** Thanks for your suggestion, we will re-organize the background part into preliminary and related work to include more current work in our main text. For more detailed comparison between the previous literature with our work, we would kindly refer to our Appendix A.
>
> > **Q4:** In tables 1-4, the natural accuracy and the robustness performance in the centralized counterpart can be validated as references, and the gap in comparison to the achieved performance in federated learning framework should be reviewed so that the readers can deeply understand the limitation in this topic.
>
> **A4:** We actually have provided the centralized counterpart as the reference in Table 17, which provides the complete performance results on three benchmark datasets with both Non-IID and IID settings. Although there is a certain gap between the centralized counterpart with the Non-IID performance using different federated optimization methods, another interesting finding in our experimental setting is that our methods with advanced federated optimization methods like Scaffold can achieve the comparable or even better performance in the IID setting. As further discussed in Appendix E.10, this performance gain could be from the distributed training paradigm that helps adversarial training converge to the more robust optimum by the divide-and-conquer mechanism or the ensemble paradigm [1]. It is also an interesting future direction to investigate. We are worried about the unnecessary distraction in our main text and thus put it in Appendix E.10 and referred to it in Section 4.2.
>
> [1] Tramèr, Florian, et al. "Ensemble Adversarial Training: Attacks and Defenses." International Conference on Learning Representations. 2018.

---

> ### Author Response · Authors · 2022-11-17
> **Would you mind confirming if you have further questions? Thanks!**
>
> Dear Reviewer YWt9,
>
> We appreciate your efforts and time in providing constructive feedback and comments! We have carefully considered your advice/questions and provided as much refinement and experiments to address your concerns regarding the submission. Would you mind checking our response with the updated version, and confirming if you have further questions?
>
> Sincerely,
>
> The authors

---

> ### Author Response · Authors · 2022-11-18
> **Would you mind checking our response? Welcome for more discussions.**
>
> Dear reviewer YWt9,
>
> Thanks again for your time and efforts in reviewing our paper. As the end of Discussion Stage 1 is approaching and the window for paper revision is closing, here is a summary of our previous response and update:
>
> - Properly revised the description about the conjecture (see the revision and Section 3.1).
> - Expanded discussion about the intuition behind Eq.(1) that introduces a lower bound (see Section 4.2).
> - Re-organized the background part and separated it into related work and preliminary (see Sections 2 and 3, as well as Appendix A)
> - Revised the explanation about the problem with proper empirical verification and description (see Section 3.1 and Appendixes C and E.4).
> - Clarified the experimental arrangement in the main text to refer more results (see Section 5.2).
> - Expanded discussion about the systematic challenges in federated adversarial training (see Sections 1,2,6 and Appendixes A and F).
>
> **We humbly expect you could check our responses with our updated version, and confirm whether our response has addressed your concerns. More discussions are always welcome. Please let us know if there are any further questions or suggestions that we could clarify or improve.**
>
> Sincerely,
>
> The authors

---

> ### Author Response · Authors · 2022-11-21
> **Need further clarification? We are actively available for discussions in Discussion Stage 2.**
>
> Dear Reviewer YWt9,
>
> Thanks again for your time and valuable comments! In Discussion Stage 1, we have carefully considered your initial advice/questions and provided the individual responses (e.g., A1-A6 in the [details](https://openreview.net/forum?id=eKllxpLOOm&noteId=RpoxcEqnAq)) with the revised submission (and a [shortened summary](https://openreview.net/forum?id=eKllxpLOOm&noteId=yNuQkg0t_Nb)) based on your constructive suggestions.
>
> - Overall, considering the comments from all reviewers, we revised our claims by properly linking more supportive evidence into the main text (e.g., Figures 2/3/9/10 related to exacerbated heterogeneity and our SFAT), reorganized the background part into related work and preliminaries to expand the discussion for challenges and significance (e.g., Sections 2/6 and Appendixes A/F), further explained the intuition behind introducing the lower bound with empirical evidence (e.g., Appendix E.3/E.4), and refined our presentation of the comprehensive exploration (e.g., Section 5 and Appendixes C/E.1-E.12).
>
> **In Discussion Stage 2**, we are actively available for further clarification and discussion with you in the openreview system if there are any unclear parts or concerns/questions. **We would appreciate it if you could confirm whether you are satisfied with our response, and we will do our best to address any further questions/suggestions during the discussion phase. Thanks!**
>
> Sincerely,
>
> The authors of Paper1198

---

> ### Author Response · Authors · 2022-11-23
> **Looking forward to your responses or further suggestions/comments!**
>
> Dear Reviewer YWt9,
>
> We have carefully considered and addressed your initial concerns regarding our paper. We are happy to discuss them with you in the openreview system if you feel that there still are some concerns/questions. We also welcome new suggestions/comments from you!
>
> Best regards,
>
> The authors of Paper1198

---

> > ### Comment · Reviewer_YWt9 · 2022-11-24
> > **Thanks for the rebuttal**
> >
> > My concerns have been well addressed. Thus, I increase my score.

---

> > > ### Author Response · Authors · 2022-11-24
> > > **Thank you for the response and your support!**
> > >
> > > Dear Reviewer YWt9,
> > >
> > > Thanks a lot for your response and support! We are glad that our rebuttal well addresses your concerns.
> > >
> > > Sincerely,
> > >
> > > The authors of Paper1198

---

### Official Review · Reviewer_CYPm · 2022-10-25

**Confidence:** 2
**Correctness:** 4
**Technical Novelty And Significance:** 2
**Empirical Novelty And Significance:** 3
**Recommendation:** 6

**Clarity, Quality, Novelty And Reproducibility:**

I think the overall writing is clear and easy to follow. It would be better if the authors could provide more interpretations of the theoretical result, like how this is related to the experiment section.

**Strength And Weaknesses:**

Strengths: 1. The problem itself is interesting. This paper provides a simple and practical solution.
Weaknesses: 1.  To my understanding, this convergence guarantee is a variation of the standard federated learning, e.g. (Li et al, 2019) with an $\alpha$-slack variable. Theorem 3.2 and 3.3 are saying this algorithm is converging faster compared to FAT. This is not related to any claims about better robust accuracy or generalization, which is demonstrated through experiments.  It would be better to provide theoreical results showing better robust accuracy or provide empirical results showing faster convergence, which verifies the theoretical claim.


Question:
1. I am confused about the second figure in FIgure 5. It seems that the SFAT robust accuracy also decreases after 40 rounds. How is this figure different from Figure 1 (b)?

2. I understand that SFAT could achieve better overall robust accuracy compared to FAT. But for those clients that have large adversarial loss. My intuition is that these clients are down-weighted and would have worse performance than FAT. Is this true? It would be better to provide some experiments that verifies generalization performance through clients.

**Summary Of The Paper:**

This paper studies adversarial training under the federated learning framework. The current challenge in federated adversarial training (FAT) is that robust accuracy deteriorates in the later stage of training.  The authors believe this is due to the intensified heterogeneity caused by the inner maximization step in each client's local training step. To overcome this, the authors proposed $\alpha$-slack mechanism. To my understanding, this mechanism reweights the parameter update by its adversarial loss. The clients with lower adversarial loss is upweighted by $1+\alpha$. On the theoretical side, this paper shows the convergence property of the reweighted loss. Empirically, this paper provides extensive experiments and demonstrates the effectiveness of the proposed mechanism.

**Summary Of The Review:**

This paper provides a simple and practical approach to mediate the robustness deterioration issue induced by intensified data heterogeneity.  I am not an expert on this topic so I could not evaluate the significance of the contribution. The effectiveness of the proposed method is thoroughly studied through experiments.

---

> ### Author Response · Authors · 2022-11-12
> **Response to Reviewer CYPm [2/2]**
>
> > **Q3:** I understand that SFAT could achieve better overall robust accuracy compared to FAT. But for those clients that have large adversarial loss. My intuition is that these clients are down-weighted and would have worse performance than FAT. Is this true? It would be better to provide some experiments that verifies generalization performance through clients.
>
>  **A3:** Thank you for the question. In Figure 3, we can find there is no dominant client exists, which means that all clients are ever chosen to be up-weighted/down-weighted during the training. Thus, SFAT explores to help all clients to be better. Besides, the global model is redistributed to each client in each communication round of federated learning, which can also avoid the bias accumulation. To verify each client's performance in our experiments, we report the robust training accuracy of each client in our experiments as follows. The results show the training performance of each client is not worse than the original FAT.
>
> **Table 2.** Robust training accuracy on each client w.r.t. different methods in our original experiments.
>
> |          |        |            |            |   Client   |            |            |
> | :------: | :----: | :--------: | :--------: | :--------: | :--------: | :--------: |
> | Dataset  | Method |     1      |     2      |     3      |     4      |     5      |
> | CIFAR-10 |  FAT   |   59.35%   |   65.97%   |   75.96%   |   77.12%   |   80.37%   |
> | CIFAR-10 |  SFAT  | **60.05%** | **66.57%** | **77.17%** | **78.79%** | **83.23%** |
> |   SVHN   |  FAT   |   89.50%   |   82.12%   |   89.60%   |   81.96%   |   87.88%   |
> |   SVHN   |  SFAT  | **91.69%** | **85.58%** | **91.93%** | **85.31%** | **90.35%** |
>
> To further check the generalization performance of each client, we also conducted an extra experiment where we split a small part of local data in each client to serve as the local test data and evaluate their training and test performance using FAT and the proposed SFAT on the SVHN dataset. According to the following table, the generalization performance (indicated by the gap of robust training and test accuracy) of SFAT is also generally better than FAT.
>
> **Table 3.** Robust training and test accuracy on each client w.r.t. different methods.
>
> |      | Setting |       |        |        | Client |        |        |
> | :--: | :-----: | :---: | :----: | :----: | :----: | :----: | :----: |
> |      | Method  |       |   1    |   2    |   3    |   4    |   5    |
> | SVHN |   FAT   | Train | 87.51% | 88.19% | 85.91% | 78.99% | 79.17% |
> | SVHN |   FAT   | Test  | 72.96% | 70.64% | 64.30% | 61.97% | 66.57% |
> | SVHN |   FAT   |  Gap  | 14.55% | 17.55% | 21.61% | 17.02% | 12.65% |
> | SVHN |  SFAT   | Train | 87.59% | 88.41% | 85.97% | 79.27% | 79.04% |
> | SVHN |  SFAT   | Test  | 73.65% | 71.46% | 65.80% | 63.48% | 66.62% |
> | SVHN |  SFAT   |  Gap  | 13.94% | 16.95% | 20.17% | 15.79% | 12.42% |

---

> ### Author Response · Authors · 2022-11-12
> **Response to Reviewer CYPm [1/2]**
>
> Thank you for your time devoted to reviewing this paper and your constructive suggestions. Here are our detailed replies to your questions.
>
> > **Q1:** It would be better to provide theoreical results showing better robust accuracy or provide empirical results showing faster convergence, which verifies the theoretical claim.
>
> **A1:** Thank you for the constructive advice. Here, we provide the empirical verification about the theoretical claim by tracing the training loss. In the following table, we show that our $\alpha$-slack mechanism can achieve a faster convergence as the training loss quickly achieves to stable than FAT during the same training epochs. We will add the verification as well as the corresponding discussion in the draft and claim the empirical support in the theoretical analysis.
>
> **Table 1.** Empirical verification on CIFAR-10 datasets for $E|\cdot|$ in Theorem 3.2.
>
> | Method |    10    |    20    |    30    |    40    |    50    |    60    |    70    |    80    |    90    |   100    |
> | :----: | :------: | :------: | :------: | :------: | :------: | :------: | :------: | :------: | :------: | :------: |
> |  FAT   |   1.87   |   1.74   |   1.64   |   1.56   |   1.49   |   1.44   |   1.39   |   1.35   |   1.32   |   1.29   |
> |  SFAT  | **1.83** | **1.69** | **1.59** | **1.51** | **1.44** | **1.39** | **1.35** | **1.30** | **1.27** | **1.24** |
>
> **Table 2.** Empirical verification on SVHN datasets for $E|\mathcal{L}|-\mathcal{L}^*$ in Theorem 3.3.
>
> | Method |    10    |    20    |    30    |    40    |    50    |    60    |    70    |    80    |    90    |   100    |
> | :----: | :------: | :------: | :------: | :------: | :------: | :------: | :------: | :------: | :------: | :------: |
> |  FAT   |   0.37   |   0.29   |   0.23   |   0.19   |   0.15   |   0.12   |   0.09   |   0.08   |   0.06   |   0.04   |
> |  SFAT  | **0.22** | **0.15** | **0.11** | **0.08** | **0.05** | **0.04** | **0.02** | **0.02** | **0.01** | **0.01** |
>
>
> > **Q2:** I am confused about the second figure in FIgure 5. It seems that the SFAT robust accuracy also decreases after 40 rounds. How is this figure different from Figure 1 (b)?
>
> **A2:** The experiment in the second figure of Figure 5 ($1+\alpha/1-\alpha$=1.2) is to show the improvement of SFAT over FedAvg and FedProx, and does not incorporate a better $\alpha$ setting, different from that in Figure 1 (b) ($1+\alpha/1-\alpha$=1.6). In Figure 1 (b), we adopted a stronger $\alpha$-slack with a larger $\alpha$ value. We will add the corresponding experimental setup in the appropriate position to avoid this confusion. For the ablation study on $\alpha$ in our experiments, please refer to Figure 11, where we conduct a comprehensive ablation to show the effects of $\alpha$.

---

> ### Author Response · Authors · 2022-11-17
> **Would you mind confirming if you have further questions? Thanks!**
>
> Dear Reviewer CYPm,
>
> We appreciate your efforts and time in providing constructive feedback and comments! We have carefully considered your advice/questions and provided as much refinement and experiments to address your concerns regarding the submission. Would you mind checking our response with the updated version, and confirming if you have further questions?
>
> Sincerely,
>
> The authors

---

> ### Author Response · Authors · 2022-11-18
> **Would you mind checking our response? Welcome for more discussions.**
>
> Dear reviewer CYPm,
>
> Thanks again for your time and efforts in reviewing our paper. As the end of Discussion Stage 1 is approaching and the window for paper revision is closing, here is a summary of our previous response and update:
>
> - Added empirical verification about the theoretical analysis (see the revision and Appendix E.7).
> - Clarified the different setups in the experiment section and mentioned the comprehensive ablation about the $\alpha$ in the main text (see Section 5.1 and Appendix E.4).
> - Added further investigation and discussion on the robust performance of each client (see Appendix E.1).
>
> **We humbly expect you could check our responses with our updated version, and confirm whether our response has addressed your concerns. More discussions are always welcome. Please let us know if there are any further questions or suggestions that we could clarify or improve.**
>
> Sincerely,
>
> The authors

---

> ### Author Response · Authors · 2022-11-21
> **Need further clarification? We are actively available for discussions in Discussion Stage 2.**
>
> Dear Reviewer CYPm,
>
> Thanks again for your time and valuable comments! In Discussion Stage 1, we have carefully considered your initial advice/questions and provided the individual responses (e.g., A1-A3 in the [details](https://openreview.net/forum?id=eKllxpLOOm&noteId=Pfz8ipcYU1Q)) with the revised submission (and a [shortened summary](https://openreview.net/forum?id=eKllxpLOOm&noteId=yNuQkg0t_Nb)) based on your constructive suggestions.
>
> - Overall, considering the comments from all reviewers, we revised our claims by properly linking more supportive evidence into the main text (e.g., Figures 2/3/9/10 related to exacerbated heterogeneity and our SFAT), reorganized the background part into related work and preliminaries to expand the discussion for challenges and significance (e.g., Sections 2/6 and Appendixes A/F), further explained the intuition behind introducing the lower bound with empirical evidence (e.g., Appendix E.3/E.4), and refined our presentation of the comprehensive exploration (e.g., Section 5 and Appendixes C/E.1-E.12).
>
> **In Discussion Stage 2**, we are actively available for further clarification and discussion with you in the openreview system if there are any unclear parts or concerns/questions. **We would appreciate it if you could confirm whether you are satisfied with our response, and we will do our best to address any further questions/suggestions during the discussion phase. Thanks!**
>
> Sincerely,
>
> The authors of Paper1198

---

> ### Author Response · Authors · 2022-11-23
> **Looking forward to your responses or further suggestions/comments!**
>
> Dear Reviewer CYPm,
>
> We have carefully considered and addressed your initial concerns regarding our paper. We are happy to discuss them with you in the openreview system if you feel that there still are some concerns/questions. We also welcome new suggestions/comments from you!
>
> Best regards,
>
> The authors of Paper1198

---

### Official Review · Reviewer_JVxX · 2022-10-25

**Confidence:** 4
**Correctness:** 3
**Technical Novelty And Significance:** 2
**Empirical Novelty And Significance:** 3
**Recommendation:** 6

**Clarity, Quality, Novelty And Reproducibility:**

The clarity and quality of this paper is great. For the novelty please refer to my points in the Weaknesses part. Reproducibility is not applicable for now since the authors have not provided the source codes, while they promised to do so in the reviewing phase. I will update then. However, I think the proposed is simple enough and there should be no problem in reproducing the results.

**Strength And Weaknesses:**

Strength of this work is primally empirical. Firstly it seems to be the first empirical breakthrough in terms of combining FL and adversarial training, with plenty of supporting experiments showing the effectiveness of the proposed method. Secondly, it provides plenty of interesting and inviting observations in multiple dimensions, which I believe would be of interest for the community.

However, the technical novelty issue is critical. And the core of the novelty issue is that the proposed method looks simply a modified aggregation method that has nothing related to the adversarial training objective. It perfectly applies to normal FL with only natural samples. Let me be more specific.

From the perspective of the motivation, that is discussed in Sec 3.1, it is purely a conjecture/argument and there is neither qualitative analysis or quantitative evidence that actually supports the statement that the local adversarial optimization process indeed "exacerbate" the heterogeneity.

From the perspective of the algorithm, as I said above, the proposed method simply upweights updates from clients that have smaller local (adversarial) objective. It does not matter if the local objectives are adversarial or not. And the final form of the algorithm is *very* similar to the **FedSoftBetter** aggregation method proposed in [1], which reweights updates from clients using a slightly different metric but also depends on the local objective magnitude. The existence of this prior work, in my opinion, significantly undermines the technical contribution of this work. However, I do recognize that [1] is quite new and is only a preprint work. I am not sure how it should influence the evaluation of this submission. I would love to invite comments from other reviewers and discussion from the authors on it.

From the perspective of empirical results, the proposed method seems to be most beneficial in the natural cases in every experimental setting presented in the paper (in terms of final accuracy improvements). This observation further convinces my points above that the proposed method is not specific to adversarial training at all.

Lastly, which is more of a constructive comment instead of criticism, the authors only consider constant alpha in this work, which for sure changes the original global adversarial objective. A natural idea is to gradually decrease alpha during FL so that the objective would asymptotically get closer to the real objective. However, this is empirically contrary to the observation that FAT degrades at the later stage of training. So the investigation on this question is worthwhile and interesting.


UPDATE: sorry that I forgot to attach the reference above.
[1] Mansour, Adnan Ben, et al. "Federated Learning Aggregation: New Robust Algorithms with Guarantees." arXiv preprint arXiv:2205.10864 (2022).

**Summary Of The Paper:**

This paper tries to fix the failure of the direct combination of federated learning (FL) and adversarial training, which is indicated by the deterioration of the adversarial accuracy at the later stage of FL. The authors proposed the so-called "alpha-slack" mechanism that upweights the clients with smaller (local) adversarial losses (and downweights the clients with larger losses) during aggregation in the cloud. The authors also show that the proposed mechanism asymptotically converges to the optimal solution of the modified adversarial objective. Extensive experiments are presented to show the effectiveness of the proposed method.

**Summary Of The Review:**

This work is empirically solid but technically weak because of the clear disconnection of the proposed method and the adversarial setting AND the prior work (FedSoftBetter) which is very similar algorithmically.

====

Update: after reading the authors' responses to my review and their discussions with other reviewers, I found my concerns have been mostly addressed. Therefore, I raised my score from 5 to 6.

---

> ### Author Response · Authors · 2022-11-12
> **Response to Reviewer JVxX [3/3]**
>
> > **Q4:** Lastly, which is more of a constructive comment instead of criticism, the authors only consider constant alpha in this work, which for sure changes the original global adversarial objective. A natural idea is to gradually decrease alpha during FL so that the objective would asymptotically get closer to the real objective. However, this is empirically contrary to the observation that FAT degrades at the later stage of training. So the investigation on this question is worthwhile and interesting.
>
> **A4:** Thanks for the constructive suggestion. In the following, we present the experimental results that progressively anneal the coefficient to vanilla federated adversarial training during the training and compare with SFAT on CIFAR-10 and SVHN. According to the results, gradually annealing-$\alpha$ SFAT achieves slightly lower robust accuracy than the constant-$\alpha$ SFAT and sometimes may introduce the large drop of natural accuracy (e.g., on CIFAR-10). It indicates that keeping a consistent $\alpha$ for the $\alpha$-slack may be a better choice. On the other hand, it also confirms the intuition that using a gradually decreased alpha is empirically contrary to the observation of robustness deterioration (or the heterogeneity exacerbation) at the later stage of training.
>
> **Table 2.** Comparison using different $\alpha$ schedules on CIFAR-10 and SVHN datasets.
>
> | Dataset  | $\alpha$-slack                                | Methods | Natural    | FGSM       | PGD-20     | CW$_{\infty}$ |
> | -------- | --------------------------------------------- | ------- | ---------- | ---------- | ---------- | ------------- |
> | CIFAR-10 | -                                             | FAT     | 57.45%     | 39.44%     | 32.58%     | 30.52%        |
> | CIFAR-10 | $\frac{1+\alpha}{1-\alpha}:1.4$               | SFAT    | **63.44%** | **45.13%** | **37.17%** | **33.99%**    |
> | CIFAR-10 | $\frac{1+\alpha}{1-\alpha}:1.4\rightarrow1.0$ | SFAT    | 60.63%     | 43.53%     | 36.23%     | 33.22%        |
> | SVHN     | -                                             | FAT     | 91.24%     | 87.95%     | 68.87%     | 67.39%        |
> | SVHN     | $\frac{1+\alpha}{1-\alpha}:1.2$               | SFAT    | 91.25%     | 88.28%     | **71.72%** | **69.79%**    |
> | SVHN     | $\frac{1+\alpha}{1-\alpha}:1.2\rightarrow1.0$ | SFAT    | **91.68%** | **88.55%** | 71.44%     | 69.76%        |
>
> We will add the experiments and the discussion in our appendix.
>
> > **Q5:** Reproducibility is not applicable for now since the authors have not provided the source codes, while they promised to do so in the reviewing phase.
>
> **A5:** Thank you for pointing out this. To avoid the concerns about the reproducibility and the detailed setups, we will open our source code in an anonymous repository.

---

> ### Author Response · Authors · 2022-11-12
> **Response to Reviewer JVxX [2/3]**
>
> > **Q3:** The final form of the algorithm is very similar to the FedSoftBetter aggregation method proposed in [1], which reweights updates from clients using a slightly different metric but also depends on the local objective magnitude. The existence of this prior work, in my opinion, significantly undermines the technical contribution of this work. However, I do recognize that [1] is quite new and is only a preprint work. I am not sure how it should influence the evaluation of this submission. I would love to invite comments from other reviewers and discussion from the authors on it.
>
> **A3:** Thank you for recommending new preprint work [1] to us. We will add it into the draft and discuss the difference between this work and ours. In the following, we would like to clarify two points:
>
> 1) SFAT and FedSoftBetter have different motivations and underlying principles, since SFAT focuses on federated adversarial training and origins from our $\alpha$-slack mechanism for alleviating the exacerbated heterogeneity, while FedSoftBetter focuses on ordinary federated learning and originates as a specific strategy for federated aggregation from the theoretical analysis on convergence bound. Note that as discussed in Q1, SFAT is not for ordinary federated learning.
>
> 2) In practice, the technical forms of the two weighting mechanisms are different. SFAT directly ranks the losses of all clients and constructs the weights, while the weights of FedSoftbetter build upon the gap between the client loss and the client optimal loss. To empirically verify the difference between FedSoftBetter and our SFAT, we conduct the experiments on CIFAR-10 and SVHN to compare their performance in the following table. The results show that its empirical performance is slightly better than FedAvg but not better than our SFAT.
>
> **Table 1.** Comparison using different methods on CIFAR-10 and SVHN datasets.
>
> | Dataset  | Methods       | Natural    | FGSM       | PGD-20     | CW$_{\infty}$ |
> | -------- | ------------- | ---------- | ---------- | ---------- | ------------- |
> | CIFAR-10 | FAT           | 57.45%     | 39.44%     | 32.58%     | 30.52%        |
> | CIFAR-10 | FedSoftBetter | 58.86%     | 40.23%     | 32.78%     | 30.76%        |
> | CIFAR-10 | SFAT          | **63.44%** | **45.13%** | **37.17%** | **33.99%**    |
> | SVHN     | FAT           | 91.24%     | 87.95%     | 68.87%     | 67.39%        |
> | SVHN     | FedSoftBetter | **91.56%** | 88.13%     | 68.80%     | 67.83%        |
> | SVHN     | SFAT          | 91.25%     | **88.28%** | **71.72%** | **69.79%**    |
>
> [1] Mansour, Adnan Ben, et al. "Federated Learning Aggregation: New Robust Algorithms with Guarantees." arXiv preprint arXiv:2205.10864 (2022).

---

> ### Author Response · Authors · 2022-11-12
> **Response to Reviewer JVxX [1/3]**
>
> Thank you for your time devoted to reviewing this paper and your constructive suggestions. Here are our detailed replies to your questions.
>
> > **Q1:** However, the technical novelty issue is critical. And the core of the novelty issue is that the proposed method looks simply a modified aggregation method that has nothing related to the adversarial training objective. It perfectly applies to normal FL with only natural samples. From the perspective of empirical results, the proposed method seems to be most beneficial in the natural cases in every experimental setting presented in the paper (in terms of final accuracy improvements). This observation further convinces my points above that the proposed method is not specific to adversarial training at all.
>
> **A1:** We are sorry that we have not clearly emphasized our specific design for federated adversarial training instead of ordinary federated learning at the beginning. Actually, in the experimental parts, we do answer this concern. As shown in the left two panels of Figure 5, our $\alpha$-Slack mechanism fails to make effects in ordinary federated learning but make positive improvements in federated adversarial training. We will add the necessary emphasis in the introduction when introducing the uniqueness of federated adversarial training and our specific design to overcome the potential problem.
>
> Regarding the performance, it is not proper to compare the improvement on natural accuracy with that on robust accuracy, since the difficulty degree of natural examples and the difficulty degree of adversarial examples can be at different levels. Besides, it is not always the case that our method is more beneficial to the natural accuracy. For example, on SVHN dataset (please refer the results in Tables 3, 7, 8, or 14), we can also find that the improvement on robust accuracy (about 2%) is larger than the improvement on natural accuracy (about 0.1%).
>
> > **Q2:** From the perspective of the motivation, that is discussed in Sec 3.1, it is purely a conjecture/argument and there is neither qualitative analysis or quantitative evidence that actually supports the statement that the local adversarial optimization process indeed "exacerbate" the heterogeneity.
>
> **A2:** The corresponding qualitative analysis is presented in Figure 6(a), where we verify the intensified heterogeneity in the perspective of client drift [1,2]. However, we appreciate the reviewer's challenge about the clarity of the motivation. We will add Figure 6(a) into Figure 2 to better improve the motivation part.
>
> Besides, the same quantitative analysis has also been conducted on our SFAT in Figures 4, 10, and 11 to compare with the original FAT, which confirms the rationality of our $\alpha$-slack mechanism and SFAT with $\alpha$-slack indeed alleviates the exacerbated heterogeneity and achieve better robust accuracy. In the corresponding parts, we will add the necessary clarification of the experimental support to facilitate the reviewers and readers to refer to the quantitative analysis.
>
> [1] Sai Praneeth Karimireddy, Satyen Kale, Mehryar Mohri, Sashank Reddi, Sebastian Stich, and Ananda Theertha Suresh. Scaffold: Stochastic controlled averaging for federated learning. In ICML, 2020.
>
> [2] Tian Li, Anit Kumar Sahu, Manzil Zaheer, Maziar Sanjabi, Ameet Talwalkar, and Virginia Smith Federated optimization in heterogeneous networks. In MLSys, 2018.

---

> ### Author Response · Authors · 2022-11-17
> **Would you mind confirming if you have further questions? Thanks!**
>
> Dear Reviewer JVxX,
>
> We appreciate your efforts and time in providing constructive feedback and comments. We have carefully considered your concerns/advice and provided as much refinement and experiments to address them regarding the submission. Would you mind checking our response and confirming whether we have solved your concerns? If there are any further questions or suggestions, we will try our best to address them to improve our submission.
>
> Sincerely,
>
> The authors

---

> ### Author Response · Authors · 2022-11-18
> **Would you mind checking our response? Welcome for more discussions.**
>
> Dear reviewer JVxX,
>
> Thanks again for your time and efforts in reviewing our paper! As the end of Discussion Stage 1 is approaching and the window for paper revision is closing, here is a summary of our previous response and update:
>
> - Properly revised the motivation part and mentioned more empirical evidence (from the appendix) in the main text about exacerbated heterogeneity and the effects of our SFAT (see Section 3.1, Appendix E.3 and E.4).
> - Added detailed discussion with the newly preprint work and provided the comprehensive comparison (see Appendix A).
> - Properly emphasized that our method is tailored for federated adversarial training with supportive results in 'Non-AT v.s. AT' (see Section 5.2 and Appendix A).
> - Added results and discussion about gradually decreasing the $\alpha$ in the experiment (see Appendix E.5).
>
> **We humbly expect you could check our responses with our updated version, and confirm whether our response has addressed your concerns. More discussions are always welcome. Please let us know if there are any further questions or suggestions that we could clarify or improve.**
>
> Sincerely,
>
> The authors

---

> ### Author Response · Authors · 2022-11-21
> **Need further clarification? We are actively available for discussions in Discussion Stage 2.**
>
> Dear Reviewer JVxX,
>
> Thanks again for your time and valuable comments! In Discussion Stage 1, we have carefully considered your initial concerns/advice and provided the individual responses (e.g., A1-A5 in the [details](https://openreview.net/forum?id=eKllxpLOOm&noteId=XJPFGxSE2QZ)) with the revised submission (and a [shortened summary](https://openreview.net/forum?id=eKllxpLOOm&noteId=yNuQkg0t_Nb)) based on your constructive suggestions.
>
> - Overall, considering the comments from all reviewers, we revised our claims by properly linking more supportive evidence into the main text (e.g., Figures 2/3/9/10 related to exacerbated heterogeneity and our SFAT), reorganized the background part into related work and preliminaries to expand the discussion for challenges and significance (e.g., Sections 2/6 and Appendixes A/F), further explained the intuition behind introducing the lower bound with empirical evidence (e.g., Appendix E.3/E.4), and refined our presentation of the comprehensive exploration (e.g., Section 5 and Appendixes C/E.1-E.12).
>
> **In Discussion Stage 2**, we are actively available for further clarification and discussion with you in the openreview system if there are any unclear parts or concerns/questions. **We would appreciate it if you could confirm whether you are satisfied with our response, and we will do our best to address any further questions/suggestions during the discussion phase. Thanks!**
>
> Sincerely,
>
> The authors of Paper1198

---

> ### Author Response · Authors · 2022-11-23
> **Looking forward to your responses or further suggestions/comments!**
>
> Dear Reviewer JVxX,
>
> We have carefully considered and addressed your initial concerns regarding our paper. We are happy to discuss them with you in the openreview system if you feel that there still are some concerns/questions. We also welcome new suggestions/comments from you!
>
> Best regards,
>
> The authors of Paper1198

---

> > ### Comment · Reviewer_JVxX · 2022-11-23
> > **Thank you for the responses**
> >
> > I sincerely appreciate the authors' efforts and responses. After reading all the feedbacks and discussions, my concerns are mostly addressed and hence I raise my score to 6.

---

> > > ### Author Response · Authors · 2022-11-24
> > > **Thank you for the response and support!**
> > >
> > > Dear Reviewer JVxX,
> > >
> > > We are glad that our responses and revision address your concerns. Thanks a lot for reading our responses and raising the score!
> > >
> > > Sincerely,
> > >
> > > The authors of Paper1198

---

### Author Response · Authors · 2022-11-17
**General response -- thanks to all reviewers for the valuable feedback**


We are very glad to see that the reviewers find our work **impressive**, **interesting**, **empirical solid with great clarity and quality** (R1, R2, R3), and the method is **effective and practical** (R1, R2, R3), with **theoretical analysis** (R2, R3), and the experiments are **extensive and in multiple dimensions** (R1, R2, R3) and can provide **sufficient evidence** (R3) and **plenty of observations would be interest for the community** (R1).

We have addressed the reviewers' comments and concerns in **individual responses to each reviewer**. The reviews allowed us to improve our draft, and the changes made in our revised draft are summarized below:

From Reviewer JVxX

- Properly revised the motivation part and mentioned more empirical evidence (from the appendix) in the main text about exacerbated heterogeneity and the effects of our SFAT (see Section 3.1, Appendix E.3 and E.4).
- Added detailed discussion with the newly preprint work and provided the comprehensive comparison (see Appendix A).
- Properly emphasized that our method is tailored for federated adversarial training with supportive results in 'Non-AT v.s. AT' (see Section 5.2 and Appendix A).
- Added results and discussion about gradually decreasing the $\alpha$ in the experiment (see Appendix E.5).



From Reviewer CYPm

- Added empirical verification about the theoretical analysis (see the revision and Appendix E.7).
- Clarified the different setups in the experiment section and mentioned the comprehensive ablation about the $\alpha$ in the main text (see Section 5.1 and Appendix E.4).
- Added further investigation and discussion on the robust performance of each client (see Appendix E.1).


From Reviewer YWt9

- Properly revised the description about the conjecture (see the revision and Section 3.1).
- Expanded discussion about the intuition behind Eq.(1) that introduces a lower bound (see Section 4.2).
- Re-organized the background part and separated it into related work and preliminary (see Sections 2 and 3, as well as Appendix A)
- Revised the explanation about the problem with proper empirical verification and description (see Section 3.1 and Appendixes C and E.4).
- Clarified the experimental arrangement in the main text to refer to more results (see Section 5.2).
- Expanded discussion about the systematic challenges in federated adversarial training (see Sections 1,2,6 and Appendixes A and F).


**We appreciate all reviewers' comments and time again!** We have revised the paper following the suggestions and uploaded the latest version. **Would you mind checking it and confirming if you have further questions?** We will try our best to address them.

---

### Decision · Program_Chairs · 2023-01-20

**Decision:**

Accept: poster

**Justification For Why Not Higher Score:**

The problem studied is slightly niche.

**Justification For Why Not Lower Score:**

No ground to recommend negatively - it's a clear accept case

**Metareview: Summary, Strengths And Weaknesses:**

This paper aims to mitigate the loss of adversarial accuracy at the later stage of FL, when directly combing federated learning (FL) and adversarial training. The authors proposed the so-called "alpha-slack" mechanism that upweights the clients with smaller (local) adversarial losses and vice versa during aggregation in the cloud. They demonstrated the proposed mechanism asymptotically converges to the optimal solution of the modified adversarial objective. Extensive experiments are reported.

Reviewers initially raised concerns regarding the clarity of motivation, the comparison to several prior arts, and a few more ablation studies. The authors provide a robust rebuttal that seems to satisfy everybody, leading to a positive consensus of scores. AC agrees and sees this paper a clear accept case.

**Note From Pc:**

if the above contains the word "oral" or "spotlight" please see: "oral" presentation means -> notable-top-5% and "spotlight" means -> notable-top-25%. As stated in our emails, we are disassociating presentation type from AC recommendations